# Long-term effects of working memory retrieval from prioritized and deprioritized states
Frieda Born [1,2,3,4] ✉ & Bernhard Spitzer [2,3] ✉

Which factors determine whether information temporarily held in working memory (WM) can later be remembered from long-term memory (LTM)? Previous work has shown that retrieving ("testing") memories from LTM can benefit their future LTM recall. Here, we examined the extent to which a benefit for subsequent LTM may also occur after retrieval from WM, depending on whether the WM contents were retrieved from a prioritized or deprioritized state. In three experiments ($n = 383$ participants), we combined variants of a visual WM paradigm with a subsequent surprise LTM recall test. We found a LTM benefit of WM testing both for prioritized and deprioritized WM contents, which, interestingly, was stronger for the deprioritized information. This pattern showed similarly across experiments with different priority manipulations. Subsequent LTM benefits generally occurred after WM testing with a recall-like test format (continuous report), but not after simple WM comparisons against a probe. The surprisingly larger LTM benefit for deprioritized WM contents may reflect enhanced encoding of the participants' own subjective WM report – as opposed to the originally presented sample information – into LTM.

Of the myriads of information we experience in our daily lives, only a small fraction can later still be remembered. It is commonly assumed that momentary experiences are processed in Working Memory (WM), a strictly capacity-limited system that maintains information only as long as is needed for immediately upcoming tasks[1–4]. While the contents of WM typically persist only for seconds or less, some of the information may later still be retrieved from long-term memory (LTM) and potentially even persist for a lifetime[5,6]. Which factors determine whether WM contents become durable in LTM or are eventually forgotten? Previous research has shown that LTM formation is facilitated, for instance, by "deep" semantic encoding of the stimulus materials (Levels of Processing[7]), by their emotional salience (e.g., flashbulb memories[8]), and by directing top-down attention to them[9,10]. Other work has established that LTM storage is furthermore consolidated by "testing", that is, by attempting to remember ("retrieval practice") the information again at a later point in time[11–13]. From a WM perspective, these factors pertain to how information is encoded into WM, and whether the information is encoded from the environment or generated internally (the latter typically benefits subsequent LTM; for review, see ref. 14).

Beyond encoding, WM function is thought to include purposeful maintenance, updating, and retrieval processes (e.g. refs. 15–17, for review, see ref. 18. Whether and how specific (sub)processes within WM affect subsequent LTM is subject to ongoing research. Focusing on maintenance, several studies with verbal materials found that longer WM retention periods were associated with improved subsequent LTM recall ([19–21], for review, see ref. 22). These findings suggest an LTM benefit of subvocal rehearsal in WM, although similar results have been reported for non-verbal (e.g., visual) materials as well[22]. A number of studies have also investigated how subsequent LTM is affected by attentional (de-)prioritization of WM contents during maintenance, e.g., via retrospective cueing (retro-cues, see ref. 23). While many of these studies found that prioritization improved subsequent LTM[24–28], others found no such effect[29–31] or even found superior LTM when attention was diverted from the WM information[32]. Interestingly, the latter result appears consistent with a body of work by McCabe and colleagues[33–35], which showed that intermittent distraction during word list learning ("complex span" task) impaired the words' immediate WM recall, but paradoxically improved their subsequent LTM recall. This counterintuitive finding has been explained in terms of 'covert retrieval' of the WM items back into the focus of attention[33,35], based on the idea that during distraction, WM information was temporarily maintained in 'activated long-term memory' ([16,36], see also refs. 37–39). A possible corollary of this view is that retrieval processes in WM might foster subsequent LTM in quite similar ways as the well-established testing effects in the LTM

[1]Machine Learning Group, Technical University, Berlin, Germany. [2]Chair of Biopsychology, Technical University, Dresden, Germany. [3]Research Group Adaptive Memory and Decision Making, Max Planck Institute for Human Development, Berlin, Germany. [4]BIFOLD, Berlin Institute for the Foundations of Learning and Data, Berlin, Germany. ✉e-mail: f.born@tu-berlin.de; bernhard.spitzer@tu-dresden.de

literature, i.e., through retrieval from an LTM-like storage format, especially when the WM information was unattended.

However, compared to the classic testing effects in the LTM literature, the long-term consequences of overt WM testing have thus far received relatively less attention (but see refs. [40–42].). One reason for this might be that WM testing typically involves some form of reexposure to the WM information when it is presented as a recognition probe (e.g., in delayed-match-to-sample tasks) or when it is reproduced by the participant themselves (e.g., in WM recall). An overall LTM benefit of WM testing might thus be trivially expected if the WM test provides an additional learning opportunity for subsequent LTM. Studies of the well-known testing effect in the LTM literature typically control for reexposure by including matched "restudy" conditions, in which no retrieval is required[12]. Creating similar conditions in a WM-task context can be difficult because without the expectation of a WM test, there might be no reason for participants to engage in active WM maintenance ([6,43], but see also ref. [44]). On the other hand, the abovementioned 'McCabe effect' on subsequent LTM has in a few studies also been observed in trials where overt WM testing was omitted[35,45]. Together, while the idea that LTM may benefit from 'covert' WM retrieval is increasingly established, less is known about whether and how LTM is affected by overt WM retrieval during explicit WM testing.

Here, we used an alternative approach to investigate how active retrieval from WM ("WM testing") affects the longer-term memorability of information in LTM, depending on whether the information was retrieved from a prioritized or a deprioritized WM state. While previous studies found no effect of overt WM testing on the 'McCabe' effect with word lists[35,45], our WM tasks required participants to maintain visual information, specifically, the orientations of one or two rotated objects. Further, whereas previous studies of the long-term consequences of attentional (de-)prioritization often used recognition tests to probe WM (e.g., refs. [25,28,29]), we asked participants to provide continuous orientation reports, both as WM tests (Experiments 1 & 2) and when probing participants' subsequent LTM in a later surprise test. We hypothesized that continuous reporting, where participants are asked to reproduce the previously seen WM sample orientation from scratch, would promote active WM retrieval, and hence a sizable subsequent LTM benefit. Further, the continuous WM reports produced by the participants enabled us to examine whether subsequent LTM recall was biased towards (or away from) these self-generated orientations and whether such WM-based "generation effect" would depend on the attentional state of the WM information (prioritized vs. deprioritized). We manipulated the attentional state of WM information via testing order (Experiment 1) and retro-cues (Experiment 2). Lastly, in comparisons between experiments (see Experiment 3), we asked whether the LTM consequences of WM retrieval indeed depended on the format of WM testing (continuous report vs. delayed comparison).

## Methods
### Experiment 1
**Participants.** Participants ($n = 199$, 58 female, 130 male, mean age = 26.99; missing demographic information for n = 11) were recruited online via Prolific Academic (https://www.prolific.ac/). Demographic information was self-reported. No data on race or ethnicity were collected. All participants provided informed consent prior to participation, with consent obtained electronically via the Qualtrics platform (https://www.qualtrics.com). The eligibility criteria were that participants had to be between 18 and 35 years old, fluent in English, have a normal or corrected-to-normal vision, and have a minimum approval rate of 95% on Prolific. The experiment lasted approximately 40 min. Participants were reimbursed with £6.75 for completing the experiment. Partial payments were made if the experiment was not completed due to technical issues (n = 4), failed attention checks ($n = 5$), or early termination by the participant ($n = 2$). One participant (n = 1) was excluded post-experimentally for failing to perform significantly above chance in the WM task (p < 0.05, t-test against 90° angular error, one-tailed). Thus, $n = 187$ participants remained for analysis (55 female, 121 male; mean age

27.2 years; missing demographic information for $n = 11$). The experiment was approved by the Internal Review Board (IRB) of the Max Planck Institute for Human Development.

**Stimuli.** The experimental stimuli consisted of 110 pictures of animate and inanimate objects. For each participant, 100 pictures were randomly selected for the experiment, while 10 were designated for practice trials. An additional 3 pictures, identical across all participants, were used for instruction. All pictures were selected from the Bank of Standardized Stimuli (BOSS) database ([46]; licensed under CC-BY-SA 3.0). The stimuli were presented rotated in the experiment (Fig. 1). The WM-sample orientations were selected randomly and independently from a set of 16 equidistant angles from 11.25° to 348.75° in steps of 22.5°, which excluded the cardinal axes (0°, 90°, 180°, and 270°).

**Task(s).** The experiment consisted of 3 phases: a WM task (60 trials), a distractor task (approx. 1 min), and a surprise LTM test (100 trials). In the WM task, participants were asked to briefly remember the orientation of one or two WM samples. Each trial started with a fixation cross (2 s) at the screen center. In one-sample trials, the WM sample was presented for 1.5 s followed by a delay period (empty screen of 4.5 s, after which the participants were asked to remember the WM sample's orientation (see below). In two-sample trials, two WM samples were sequentially presented (with a 1 s inter-stimulus interval during which the fixation cross was shown). After a WM delay of 2 s, the orientation of one of the two objects (randomly selected) was probed (Test 1). After Test 1, in 50% of the two-sample trials (randomly varied), the experiment continued with the next trial. On the remaining two-sample trials, Test 1 was followed by another delay period (1.5 s) and participants were probed to also remember the orientation of the other, previously unprobed sample (Test 2). The assignment of sample stimuli to the one-sample, Test 1, and Test 2 conditions was counterbalanced across participants using a Latin square approach, such that on average, the exact same sample stimuli were used in each of these conditions. In all WM tests, the sample orientation in question was probed by the WM object reappearing in a new orientation (random, but at least 22.5° different from the original orientation). Participants were asked to re-rotate the probe to the remembered orientation using the left and right arrow keys (continuous report; Fig. 1) and to submit the result by pressing the space key. Trials in which participants failed to submit a response within a generously allotted time window (15 s) were excluded from the analysis (2.01% of trials on average; min = 0.00%, max = 7.93%).

After the WM task, participants performed a short distractor task (approximately one minute) in which they were asked to solve a series of simple math problems (e.g., $100 - 7 = ?$) using mental arithmetics and entering the solutions via the computer keyboard.

In the subsequent surprise LTM test, participants were asked to recall the orientations of each of the previously encountered WM samples again. Each test trial started with a fixation cross (1.5 s) after which one of the previous WM sample objects appeared in a new orientation (fully random) as LTM probe. Participants were asked to reproduce the objects' original orientation (i.e., the orientation it had as a WM sample), using the same response procedure as in the WM tests (continuous reporting; see above). Each WM sample was probed once (in random serial order) across the LTM test trials.

**Procedure.** Participants were given written instructions about the WM task and could practice the continuous reporting procedure (i.e., re-rotating stimuli via arrow keys) prior to the experiment. They were free to repeat the instructions until they felt confident to perform the task. Participants first performed six practice trials of the WM task (two per trial type: one-sample trials, and two-sample trials with and without Test 2). Thereafter, each participant performed a total of 60 WM trials (20 one-sample trials and 40 two-sample trials, in random serial order). After 14 WM trials, a brief attention check task was performed (6 trials). For this, a number word (e.g., "three") was presented at the screen center,

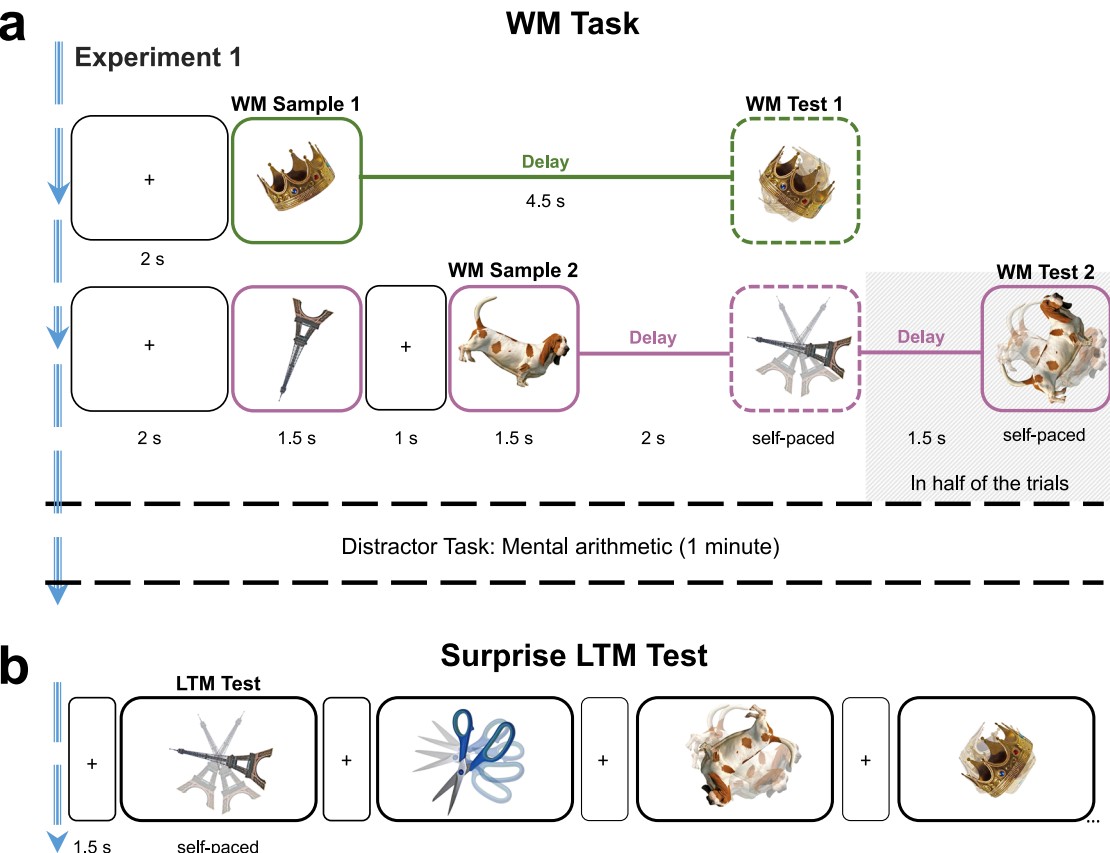

**Fig. 1 | Task layout of Experiment 1. a** WM task; examples of one-sample (top, green) and two-sample trials (bottom, purple). Participants were presented with one or two randomly oriented objects as WM Samples. When probed after the delay, the sample object reappeared in a random orientation and participants were asked to re-rotate it to its previous orientation (continuous report). In half of the two-sample trials (randomly varied), only one of the two WM samples was probed (Test 1). On the remaining two-sample trials, also the other previously unprobed WM sample was probed (Test 2). After the WM task, participants performed a short distractor task, in which they were asked to solve simple math problems. **b** Example LTM test trials. After the distractor task, participants were asked to report the orientation of all previously seen WM samples another time. Each object appeared again in a new random orientation and participants were asked to re-rotate it to the orientation it had when presented as a WM sample (see *a, left*). Stimuli were taken from the Bank of Standardized Stimuli (BOSS[46], and are licensed under CC BY–SA 3.0).

surrounded by 4 different number symbols. Participants were asked to pick the correct number symbol (e.g., "3") via arrow keys. When a participant failed this check on more than 2 of 6 trials, the experiment was aborted (see *Participants*). After completing the WM task, participants performed the distractor task (mental arithmetics, approximately 1 min). After this, they were informed about the surprise LTM test and received short instructions about its procedure. Participants then performed 100 LTM test trials in which they were asked to recall all of the sample orientations they had encountered in the WM task (including those WM samples that had not been probed in the WM task, i.e., on two-sample trials with a single WM test). Participants could take a break (self-paced for up to 2 min) after 34 trials of the WM task.

**Experiment 2**
**Participants.** For Exp. 2, we recruited another sample of n = 101 participants online (46 female, 42 male, and 1 diverse, mean age 25.3 years; missing demographic information for 12 participants), with the same modalities of recruitment, informed consent, ethics approval, reimbursement, and experiment duration as in the previous experiment. For n = 4 participants, the data was not saved due to technical problems. Two further participants were excluded due to failed attention checks, n = 1 used paper and pencil to solve the task, n = 1 started the experiment more than once, n = 3 participants did not enter any data, and n = 1 completed the task but experienced other technical problems leading to an exclusion. Thus, n = 89 participants remained for analysis (41 female, 42 male, 1 diverse; mean age 25.0 years; missing demographic information for 5 participants).

**Stimuli, task, and procedure.** The stimulus material for Exp. 2 was extended to 112 objects from the BOSS database and was otherwise identical to Exps. 1 and 3. Exp. 2 differed from the previous experiments only in WM task design. Each trial in the WM task started with the presentation of two WM samples, like the two-sample trials of the previous experiments. However, after a short delay (0.5 s fixation cross and 0.5 s blank screen), a retro-cue ("1" or "2") was displayed (1 s) which indicated which of the two WM samples would be more likely to be probed at the WM test. The retro-cue was followed by a WM delay (4 s, empty screen), after which the WM probe appeared and participants were asked to re-rotate it using the same WM-test procedure (continuous report) as in Exp. 1. We initially tested n = 55 participants (n = 47 after exclusions) with a cue validity of 75% (i.e., in 25% of trials, the uncued sample was probed). After inspecting the preliminary WM task data, we increased the cue validity to 83.33% for the remaining n = 46 participants (n = 42 after exclusions). For counterbalancing reasons, participants in the former group performed 56 trials (with 112 sample objects) and participants in the latter group performed 48 trials (with 96 sample objects) in the WM task. Within each group, the stimulus material used in the different conditions (cued/uncued x probed/unprobed) was counterbalanced across participants using a Latin square design. Trials in which participants failed to respond within the allotted time were excluded from the analysis (M = 0.33% of trials; min = 0.00%, max = 5.36%). After the WM task, participants performed a distractor task (mental arithmetics) and a surprise LTM test analogous to Exps. 1 and 3.

## Experiment 3

**Participants.** For Exp. 3, we recruited a new sample of 155 participants online (44 female, 100 male, diverse = 1, mean age 27.4 years; missing demographic information for n = 10). The modalities of recruitment, eligibility criteria, informed consent, ethics approval, and reimbursement, were the same as in Exp. 1. For n = 5 participants, the experiment was terminated prematurely due to failed attention checks, and n = 5 participants had to be excluded due to technical problems. Of the remaining participants, n = 38 were excluded because they failed to perform above chance level in the WM task (p < 0.05, Binomial test against 60% correct responses, one-tailed), leaving n = 107 participants (27 female, 50 female, mean age 27.4 years; demographic information missing for 30 participants) for analysis.

**Stimuli, task, and procedure.** The design of Exp. 3 closely resembled Exp. 1. The main difference was that in Exp. 3, the WM tests were delayed comparisons, where the WM probe was rotated +/- 14° relative to the WM sample. Participants were asked to indicate with a single key press (right or left arrow key) whether the sample-probe difference was clockwise or counterclockwise. Given the expectedly faster WM testing procedure (compared to the continuous reports in Exp. 1), we decreased the response time window to 3 s and slightly changed the lengths of the WM delays: in two-sample trials, the first WM delay was shortened to 1 s, and the second delay was extended to 2 s. In one-sample trials, the WM delay was extended to 5 s to approximately match the time between the first sample and Test 2 in two–sample trials.

Participants could practice the binary choice test format before starting with the experiment. Trials in which participants failed to respond within the allotted time were excluded from analysis (M = 0.51% of trials, min = 0.00%, max = 8.33%). The WM task in Exp. 3 was followed by a distractor task (mental arithmetics) and a subsequent surprise LTM test, using the same procedures as in Exps. 1 and 2.

### Pruning for equivalent WM performance

To account for differences in WM performance when comparing LTM performance between conditions, in our experiments with continuous reports (Exps. 1 and 2), we used a pruning approach. For each participant, we first calculated their overall WM performance (averaged across conditions) as the target performance level for pruning. Then, within each condition, trials were ranked by WM accuracy (from lowest to highest error) and trials with extreme WM reporting error (high or low, depending on condition performance) were successively removed until the difference to the target performance level was minimized. We then repeated the LTM analysis using only the WM samples that remained after this pruning. For completeness, we also performed exploratory LTM analysis of Exp. 3, where we included only samples from WM trials in which the binary WM report was correct. However, the results from this analysis were qualitatively identical to those reported in results section of Exp. 3, which included all trials.

### Statistical analysis

Throughout the analyses of continuous report data (WM and LTM) we examined memory performance in terms of absolute angular error (or deviation) in degrees (°), where lower values indicate higher accuracy (note inverted y-axes in Figures). Inspection of the residuals indicated some deviations from normality in the WM-task data. However, given our relatively large sample sizes and the robustness of repeated-measures ANOVAs to moderate non-normality, parametric tests were used. Unless stated otherwise, all reported pairwise comparisons (t-tests) were corrected for multiple testing using the Holm-Bonferroni method. This study was not preregistered.

### Reporting summary

Further information on research design is available in the Nature Portfolio Reporting Summary linked to this article.

## Results

We report the results of three experiments in which randomly oriented pictures of real-world objects were used as sample stimuli in a WM task (Fig. 1a). After completion of the WM task, a short distractor task ensued (mental arithmetics), followed by a surprise LTM test (Fig. 1b) in which participants were asked to recall the orientations of the previously encountered WM samples (Fig. 1c).

### Experiment 1

On WM task trials in Exp. 1 (n = 187), either one or two randomly oriented sample stimuli were to be maintained over a short delay period (Fig. 1a). When probed after the delay, the sample object reappeared in a random orientation and participants were asked to re-rotate it to its previous orientation (continuous report). In half of the two-sample trials (randomly varied), only one of the two samples (randomly selected) was probed. On the remaining two-sample trials, after the first WM test (Test 1), also the orientation of the other, previously unprobed sample was probed (Test 2). Thus, participants had to maintain the orientation of both WM samples until Test 1, during which the unprobed sample can be assumed to be deprioritized for the remainder of the trial. Importantly, the continuous report procedure used for WM testing in Exp. 1 provided no information about the samples' original orientations beyond the participants' own WM reports.

### WM performance

Figure 2a shows the error (absolute angular difference from the sample orientation; note inverted y-axis) of participants' reports in the WM task. As expected, WM accuracy was significantly higher (i.e. smaller errors) on one-sample trials (M = 10.05°, SE = 0.49°) compared to two-sample trials [Test 1 and 2 combined; M = 17.42°, SE = 0.73°; t(186) = -14.53, p < 0.001, 95% CI [-8.37, -6.37], d = -1.063]. Further, in the two-sample trials, performance on Test 2 (M = 20.36°, SE = 0.96°) was significantly reduced compared to Test 1 [M = 15.95°, SE = 0.71°; t(186) = 6.50, p < 0.001, 95% CI [3.07, 5.75], d = 0.476], as was expected by deprioritization of the second tested sample during and after Test 1.

In the two-sample trials, we also examined the extent to which WM accuracy was modulated by sample position. A 2 ×2 repeated measures ANOVA with the factors Sample Position (1/2) and WM Test (1/2) showed a main effect of Sample Position [F(1,186) = 4.836, p = 0.029, η2 = 0.025, 95% CI [0.00, 1.00]], indicating that first-presented samples were remembered better ("primacy" effect), and a main effect of WM Test [F(1,186) = 39.358, p < 0.001, η2 = 0.175, 95% CI [0.10, 1.00]], reflecting the lower performance on Test 2 (see above). There also was a significant interaction between the two factors [F(1,186) = 11.369, p < 0.001, η2 = 0.058, 95% CI [0.02, 1.00]], indicating that the primacy effect was stronger on Test 2 than on Test 1 (Fig. 2a). Post-hoc tests confirmed a significant primacy effect on Test 2 [M = 18.40°, SE = 1.05° vs. M = 22.15°, SE = 1.187°; t(186) = −3.239, p = 0.001, 95% CI [−6.03, −1.46], d = -0.237], but not on Test 1 [M = 16.347, SE = 0.776 vs. M = 15.523, SE = 0.786; t(186) = 1.208, p = 0.229, 95% CI [−0.52, 2.17], d = 0.089]. Together, the results from two-sample trials are in line with earlier findings of reduced WM recall after deprioritization[47–50]. In addition, the results showed a WM "primacy" effect (e.g., refs. [51,52].), which occurred only for the second-tested (deprioritized) samples.

### LTM performance

In the subsequent surprise LTM test, the participants were asked to report the orientations of all sample objects that had been presented in the WM task, including those that were not probed in a WM test. Focusing on the probed samples, as expected, participants' LTM reports (Fig. 2b) were considerably less accurate (M = 53.95°, SE = 1.13°) than their previous reports in the WM task [t(186) = 37.28, p < 0.001, 95% CI [36.37, 40.44], d = 2.726]. LTM performance for samples from one-sample WM trials appeared descriptively better (M = 50.73°, SE = 1.37°) than for samples from two-sample trials (Test 1 and 2 combined; M = 52.31°, SE = 1.20°) but the

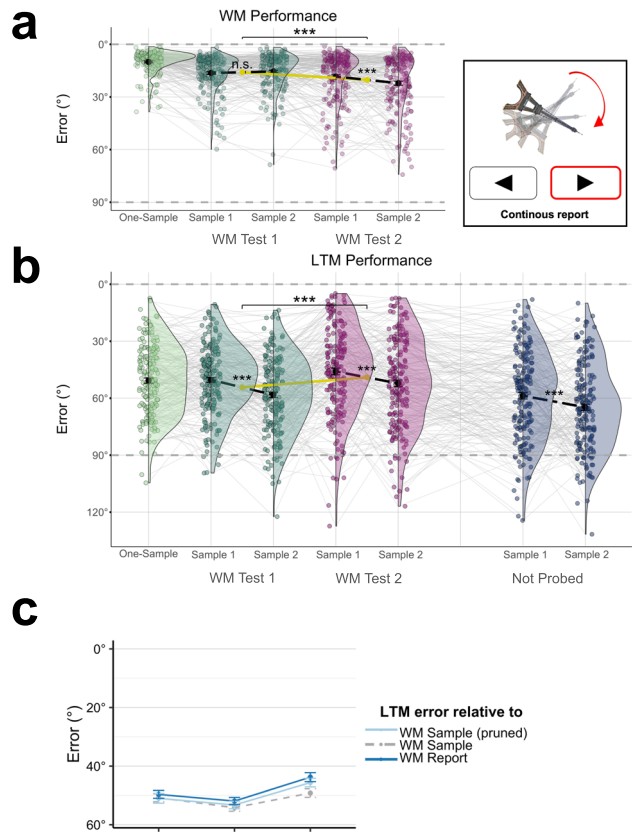

**Fig. 2 | WM and LTM performance in Experiment 1 (n = 187 participants). a** *Left*, WM task performance. WM accuracy on two-sample trials was significantly lower than on one-sample trials, and was significantly reduced after deprioritization (Test 2) compared to Test 1. Black dots, means; colored dots, individual participants. Error bars show the standard error of the mean and half-violin outlines illustrate the distribution over participants using a kernel density estimation. Asterisks on top indicate significant main effect of WM priority (p < 0.001); small asterisks below indicate significant pairwise difference (p < 0.001) between sample positions (1 or 2). Dashed horizontal lines (grey) mark ceiling (0°) and chance-level performance (90°). *Right*, in Exp. 1, a continuous report format was used in both the WM and LTM tests. Stimulus from the Bank of Standardized Stimuli (BOSS[46], licensed under CC BY–SA 3.0. **b** LTM test performance. Same plotting conventions as in *a*. In contrast to WM performance, subsequent LTM performance was *increased* for Test 2 samples compared to Test 1 samples (see main effect indicated by asterisks). **c** Light blue: LTM performance for WM samples that have been 'pruned' for equal WM performance levels across conditions (see *Results* and *Methods*). For comparison, the similarity of the LTM reports to the original WM sample orientations is shown (dashed grey), which corresponds to the LTM performance measure shown in *b*. Dark blue: similarity (in terms of absolute difference in degrees, note inverted y-axis) between the LTM- and WM-test reports. See *Results* for details.

difference was not significant [t(186) = −1.544, p = 0.124, 95% CI [-3.60, 0.44], d = -0.113]. Interestingly, focusing on the samples from two-sample trials, LTM accuracy was significantly higher for samples that had been probed second (i.e. after deprioritization) in the WM task (WM Test 2, M = 49.00°, SE = 1.48°), compared to samples that had been probed first [WM Test 1; M = 53.95°, SE = 1.20°, t(186) = −4.319, p < 0.001, 95% CI [−7.22, −2.69], d = -0.316]. Thus, whereas the WM accuracy for deprioritized samples was expectedly reduced (see WM results above), their subsequent LTM recall was surprisingly improved compared to samples from WM Test 1, and did not differ significantly from the LTM recall of samples from one-sample WM trials [see Fig. 2b; t(186) = −1.374, p = 0.171, 95% CI [−4.20, 0.75], d = −0.100].

By closer inspection, samples with the shortest distance between presentation and WM test (Sample 2, Test 1) were recalled significantly worse

in the LTM test than samples from one-sample trials [t(186) = 5.51, p < 0.01, 95% CI [0.25, 0.55], d = 0.403]. Interestingly, however, samples with the longest distance (Sample 1, Test 2), which had been retrieved from a deprioritized WM state (see Fig. 1a), were recalled significantly *better* even than the samples from one-sample WM trials [t(186) = 3.127, p = 0.002, 95% CI [0.37, 0.08], d = 0.229]. An alternative explanation for higher LTM performance for Test 2 vs. Test 1 samples could be that Test 2 was the last event in the WM trial episode, which might have rendered it more memorable, whereas Test 1 was the last event only in 50% of cases (Fig. 1). However, a control analysis showed no difference in LTM performance between Test 1 samples that were followed by a Test 2 and those that were not [i.e., where Test 1 was the last event in the WM trial; t(186) = 1.43, p = 0.155, 95% CI [−0.04, 0.25], d = 0.104], which speaks against an explanation in terms of WM-test recency.

A 2 ×2 ANOVA (specified analogously as above) of the LTM performance for the samples from two-sample WM trials showed a main effect of Sample Position [1/2; F(1,186) = 35.805, p < 0.001, η2 = 0.16, 95% CI [0.09, 1.00]], indicating better LTM recall of samples that had been presented first in the WM trial (i.e., primacy), and a main effect of Test [1/2; F(1,186) = 20.253, p < 0.001, η2 = 0.10, 95% CI [0.04, 1.00]], reflecting the improved LTM recall of deprioritized samples that had been probed on WM Test 2 (see above). There was no interaction between the two factors [F(1,186) = 0.619, p < 0.433, η2 = 0.003, 95% CI [0.00, 1.00]], and post-hoc tests showed the primacy effect on LTM performance to be significant both for samples probed first [WM Test 1; t(186) = −5.909, p < 0.001, 95% CI [−10.61, −5.30], d = −0.432] and second [WM Test 2; t(186) = −3.488, p < 0.001, 95% CI [−9.85, −2.73], d = −0.255].

We next examined for comparison the LTM performance for samples that had been presented but not probed during the WM trials. Not probed samples occurred on 50% of the two-sample trials and can be assumed to have been deprioritized after the first WM probe (WM Test 1), like those samples that had been probed on WM Test 2. The unprobed samples thus provided a baseline to quantify the LTM benefit of WM retrieval on Test 2. We indeed found that the LTM recall of the not probed orientations was significantly less accurate (M = 62.16°, SE = 1.42°) compared to those probed on WM Test 2 [t(186) = 9.745, p < 0.001, 95% CI [10.49, 15.82], d = 0.713], and also compared to those probed on the other WM Tests [WM Test 1: t(186) = 6.908, p < 0.001, 95% CI [5.86, 10.54], d = 0.505; one-sample: t(186) = 8.594, p < 0.001, 95% CI [8.81, 14.06], d = 0.628]. Thus, WM probing and/or -retrieval appeared to generally benefit subsequent LTM recall. Interestingly, the not probed samples also showed a primacy effect in LTM: those presented first in the WM trial were subsequently recalled better than those presented second [M = 58.700, SE = 1.721 vs. M = 64.80°, SE = 1.67°; t(186) = −3.31, p = 0.001, 95% CI [−9.75, −2.46], d = −0.242], just as was the case for the other (probed) samples (see above). In other words, the primacy effect on LTM performance occurred independent of WM retrieval and was more likely attributable to differences in encoding (or maintaining) the first vs. second WM sample.

### Comparing WM vs. LTM performance

To compare the WM and LTM results directly, we additionally performed a 2 × 2 × 2 repeated measures ANOVA with the factors Task (WM/LTM), WM Sample Position (1/2), and WM Test (1/2). The analysis showed anticipated main effects of Task [WM/LTM; F(1,186) = 970.059, p < 0.001, η2 = 0.839, 95% CI [0.81, 1.00]] and Sample Position [1/2; F(1,186) = 35.085, p < 0.001, η2 = 0.159, 95% CI [0.09, 1.00]], as well as a significant Task x Sample Position interaction [F(1,186) = 19.805 p < 0.001, η2 = 0.096, 95% CI [0.04, 1.00]], indicating that primacy effects were generally stronger in the LTM than in the WM tests (cf. Figures 2a and 2b). Furthermore, the Task x WM Test interaction was significant [F(1,186) = 69.396, p < 0.001, η2 = 0.272, 95% CI [0.19, 1.00]], reflecting the opposite effects of WM priority on WM vs. LTM recall performance (see above). We also found a significant three-way interaction [Task x Sample Position x WM Test; F(1,186) = 7.589 p = 0.006, η2 = 0.039, 95% CI [0.0006, 1.00]], which likely reflects the absence of primacy (or alternatively, a

recency benefit for Sample 2, see Discussion) on WM Test 1, whereas all other tests (WM and LTM) showed primacy (see Figs. 2a and 2b). No other effects were significant [WM Test, F(1,186) = 0.342, p = 0.560, η2 = 0.002, 95% CI [0.00, 1.00]; Sample Position x WM Test, F(1,186) = 1.132 p = 0.289, η2 = 0.006, 95% CI [0.00, 1.00]].

### Pruning for equivalent WM performance

Conditions that differ in WM performance (like our one-sample vs. two-sample conditions) may be expected to differ trivially also in subsequent LTM, for example, due to information loss having occurred already during WM processing. To account for this, we pruned the data post-hoc to minimize differences in WM performance between the one-sample, Test 1 and Test 2 conditions. For each participant and condition (e.g., one-sample, Test 1, and Test 2 in Exp. 1), we successively removed individual trials with extreme (high or low) WM reporting error until the WM accuracy in all conditions was maximally similar to the participant's overall mean WM accuracy (see "Methods", Pruning). We then repeated the subsequent LTM analysis using only the remaining WM samples (Fig. 2c). Pruning increased the LTM benefit of WM testing after deprioritization: we now found significantly better LTM performance for the deprioritized WM samples [Test 2; M = 46.31°, SE = 1.45°] compared even to the one-sample condition [M = 50.68°, SE = 1.369°; t(186) = −3.51, p < 0.001, 95% CI [−6.82, −1.91], d = 0.257]. Thus, after accounting for differences in WM performance, we observed even clearer LTM benefits for samples that had been retrieved from a deprioritized WM state.

### LTM recall of WM sample vs. WM report

Although participants' task in the LTM test was to recall the orientation of the originally presented WM sample (Fig. 1), their LTM reports may have been biased towards the orientations they had reported at the WM test (i.e., with WM reporting error). To examine this possibility, in the unpruned data, we inspected the similarity (in terms of absolute angular difference in °) of the LTM reports to the WM reports (Fig. 2c). In fact, the LTM reports were overall more similar to the WM reports than to the original WM sample orientations [M = 48.50°, SE = 0.812° vs. M = 51.20°, SE = 0.79°; t(186) = −10.770, p < 0.001, 95% CI [−3.28, −2.28], d = -0.788)]. This bias was evident for each of the WM-task conditions [One-sample, t(186) = −3.424, p < 0.001, 95% CI [−1.69, −0.46], d = −0.250; Test 1, t(186) = −6.101, l p < 0.001, 95% CI [−2.67, −1.37], d = −0.446] and most pronounced for the Test 2 condition [M = 44.10°, SE = 1.51°; t(186) = −8.709, p < 0.001, 95% CI [−6.42, −4.05], d = −0.637]. A repeated measures ANOVA confirmed that the increase in bias across conditions (One-sample, Test 1, Test 2) was significant [F(1,301) = 25.247, p < 0.001, η2 = 0.120, 95% CI [0.071, 1.00]]. That the bias was strongest for WM Test 2 suggests a particularly strong long-term memory of the WM-*testing* episode (i.e., of the participant's own response) after the sample information had been deprioritized. It may seem counterintuitive that in the Test 2 condition, the bias towards recalling the subjective WM reports (which include WM error) was increased in tandem with objective LTM accuracy (Fig. 2b), given that this condition showed the largest WM reporting error (Fig. 2a). However, the result can be explained when considering that the WM errors were generally much smaller than the LTM errors (cf. Fig. 2a and b). A relatively stronger bias towards the subjective WM report may thus reduce the objective LTM error to be smaller, even if the WM error was relatively larger than in other conditions.

For completeness, we also inspected whether the LTM reports were additionally biased by the (random) orientations in which the WM-test probes first appeared on screen (i.e., before the participants re-rotated them). However, the LTM reports' similarity to these probe orientations did not differ from chance level (90°) [One-sample: t(186) = -0.485, p = 1.000, 95% CI [87.81, 91.32], d = -0.35; Test 1: t(186) = 2.010, p = 0.138, 95% CI [90.02, 92.40], d = 0.147; Test 2: t(186) = 0.420, p = 1.000, 95% CI [88.67, 92.04], d = 0.031].

To summarize, while our deprioritization manipulation in Exp. 1 expectedly reduced WM-task performance, it increased the accuracy of subsequent LTM reports. The results appear consistent with a pronounced WM-"testing" effect for deprioritized materials, where participants formed a particularly strong long-term memory of the orientations they had reported at the WM test. An alternative explanation could be that the LTM performance for Test 2 items benefitted, regardless of their deprioritization, from having been maintained in WM for a longer period of time (Fig. 1a; e.g., refs. 19,21). To address this possibility, in Experiment 2, we manipulated WM priority using retro-cueing (23, for reviews, see refs. 50,53), which holds the time between sample presentation and WM test constant.

### Experiment 2

The WM task we used in Exp. 2 (n = 89) is illustrated in Fig. 3a. After the presentation of two WM samples, a visual retro-cue ("1" or "2") indicated which of the two orientations was more likely to be probed after the WM delay. The retro-cue was valid in 75% or 83.33% of the trials (see *Methods for details*). The rationale behind this manipulation was that the cued sample should be maintained with higher priority in WM, while the uncued sample (which is considerably less likely to be tested) should be deprioritized23. The WM testing procedure in Exp. 2 was otherwise identical to that in Exp. 1 (continuous reports), except that only a single item (cued or uncued) was probed on each trial. The WM task was again followed by a distractor task and a surprise LTM test analogous to Exp 1.

**WM performance.** As expected based on previous work54–56, the WM accuracy for the retro-cued orientations (M = 17.19°, SE = 1.10°) was significantly higher than for the uncued orientations [M = 22.51°, SE = 1.87°; t(88) = -2.958, p = 0.004, 95% CI [-8.89, -1.75], d = 0.314, Fig. 3b]. A 2 × 2 ANOVA with the factors Cueing (cued/uncued) and Sample Position (1/2), showed a main effect of Cueing [F(1,88) = 7.470, p = 0.008, η2 = 0.078, 95% CI [0.01, 1.00]], but no effects of Sample Position [main effect: F(1,88) = 0.165, p = 0.686, η2 = 0.002, 95% CI [0.00, 1.00]; Cueing x Sample Position: F(1,88) = 1.157, p = 0.285, η2 = 0.013, 95% CI [0.00, 1.00]]. Thus, unlike in Exp. 1, there was no significant primacy effect on WM task performance in Exp. 2 (see *Discussion*). However, the probabilistic cueing did induce the anticipated retro-cue effect, indicating that the priority manipulation was successful.

**LTM performance.** Participants' overall accuracy in the LTM test of Exp. 2 (M = 56.11°, SE = 1.101°) was at similar levels as in Exp. 1 [M = 53.95°, SE = 1.14°; t(164.08) = 1.017, p = 0.311, d = −0.119; Welch's t-test]. In Exp. 2, we again observed substantially higher LTM performance for WM samples that had been tested in the WM task (M = 51.98°, SE = 1.570), compared to unprobed samples [M = 60.23°, SE = 1.48°; t(88) = −7.51, p < 0.001, 95% CI [−11.98, −6.96], d = -0.796; Fig. 3c]. This LTM benefit of WM retrieval was evident both for cued and uncued samples [t(88) = −3.621, p = 0.002, 95% CI [−11.15, −3.25], d = -0.384 and t(88) = −4.619, p < 0.001, 95% CI [−13.31, −5.30], d = −0.487].

Turning to the LTM consequences of WM cueing, we first inspected the full data (i.e., without pruning) irrespective of the differences in WM performance between cued and uncued samples. A 2 × 2 ANOVA with the factors WM Testing (tested vs. not probed) and Cueing (cued/uncued) showed a main effect of Testing [F(1,88) = 27.974, p < 0.001, η2 = 0.241, 95% CI [0.12, 1.00]], reflecting the overall LTM benefit of WM retrieval, but no significant effects of Cueing [main effect: F(1,88) = 1.758, p = 0.188, η2 = 0.020, 95% CI [0.00, 1.00]; Testing x Cueing: F(1,88) = 0.703, p = 0.404, η2 = 0.008, 95% CI [0.00, 1.00]].

Next, we repeated the analysis after pruning the data (see Exp.1 and *Methods*) to warrant equivalent WM performance for cued and uncued samples. After pruning, the LTM results showed a significant interaction of WM Testing and Cueing [F(1,88) = 5.826, p = 0.01, η2 = 0.062, 95% CI

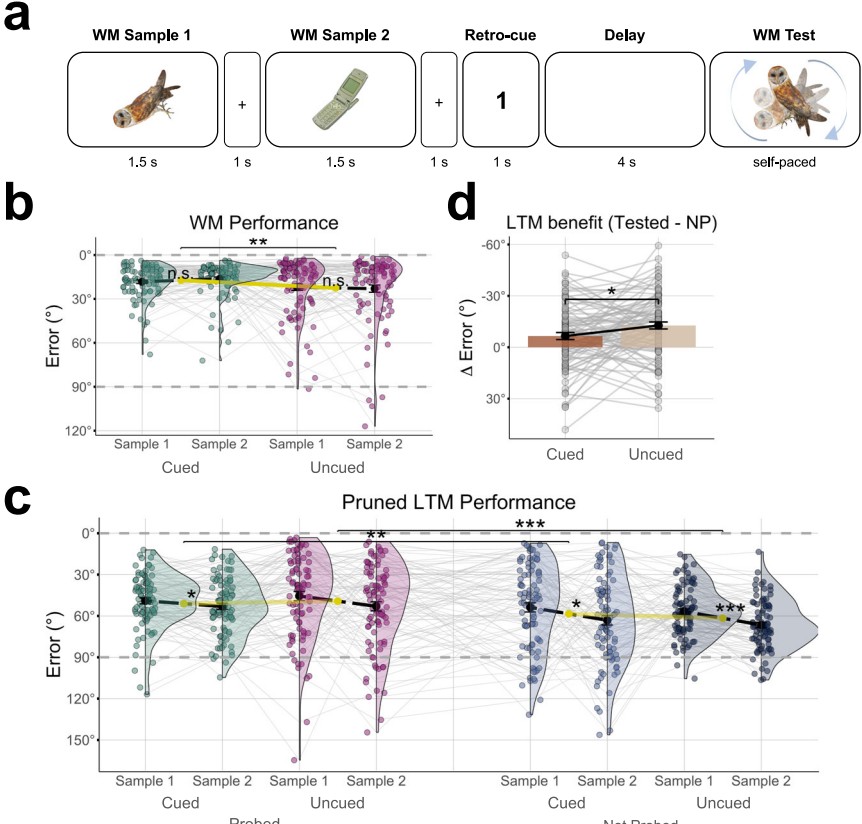

**Fig. 3 | WM and LTM performance in Experiment 2 (n = 89 participants). a** WM task. After the presentation of two randomly oriented objects (WM Sample 1 and 2), a retro-cue ("1" or "2") indicated which of the two sample orientations was most likely to be probed after the delay. In the WM test, participants were probed to recall (continuous report) the orientation of the cued sample or, in a smaller fraction of trials, the uncued sample. Images from the Bank of Standardized Stimuli (BOSS[46], licensed under CC BY–SA 3.0. **b** WM task performance (same plotting conventions as Fig. 2a). WM accuracy for the uncued information was significantly lower than for the cued information. **c** Subsequent LTM test performance after pruning for equal WM performance in the cued/uncued conditions. LTM accuracy for samples that had been probed (tested) in the WM task was significantly higher (smaller errors) than for samples that had not been probed (not probed, blue). **d** Benefit of WM retrieval (tested vs. not probed) for subsequent LTM accuracy, plotted separately for cued and uncued samples. The LTM benefit of WM retrieval was significantly larger for uncued samples.

[0.01, 1.00]; main effect of Testing: $F(1,88) = 36.164$, $p < 0.001$, $\eta2 = 0.291$, 95% CI [0.17, 1.00]; main effect of Cueing: $F(1,88) = 0.352$, $p = 0.555$, $\eta2 = 0.004$, 95% CI [0.00, 1.00]], which indicates a greater WM-testing benefit for uncued than for cued samples (see Supplementary Analysis 1 for further details). Figure 3d shows the magnitude of the WM-testing benefit (tested vs. not probed) which was significantly larger for the uncued than the cued samples. In other words, in terms of long-term memorability, deprioritized samples benefited more from WM retrieval than prioritized samples that had been retrieved (from WM) with equivalent accuracy. Further inspection of the LTM results with a 2 ×2 x 2 ANOVA (Sample Position x Testing x Cueing) showed a significant main effect of Sample Position [$F(1,88) = 24.613$, $p < 0.001$, $\eta2 = 0.219$, 95% CI [0.10, 1.00]] indicating a primacy effect on LTM recall (see also Exp. 1), but no additional new interactions [WM Testing and Cueing [$F(1,88) = 3.209$, $p = 0.07$, $\eta2 = 0.035$, 95% CI [0.00, 1.00]]; Sample Position and Cueing [$F(1,88) = 0.359$, $p = 0.551$, $\eta2 = 0.004$, 95% CI [0.00, 1.00]]; WM Testing and Sample Position [$F(1,88) = 1.255$, $p = 0.266$, $\eta2 = 0.014$, 95% CI [0.00, 1.00]]; WM Testing and Cuing and Sample Position [$F(1,88) = 0.204$, $p = 0.653$, $\eta2 = 0.002$, 95% CI [0.00, 1.00]], see also Supplementary Fig. 3].

Together, Exp. 2 confirmed a stronger LTM benefit of WM testing after deprioritization, even when the duration of WM maintenance was controlled for. While the effects of probabilistic retro-cueing were more subtle (both in terms of WM and LTM performance, Fig. 3) compared to the priority manipulation in Exp. 1 (cf. Fig. 2), they corroborate a role of WM priority for the magnitude of subsequent LTM benefits, over and above potential effects of maintenance duration.

## Experiment 3

An important aspect of the WM tests in Exp. 1 and 2 (continuous reports) was that the WM probes appeared in a quasi-random orientation (Fig. 1a, see *Methods)*, which provided no opportunity to 'restudy' the sample information. In other words, at the WM tests, participants could only possibly have ('re')studied the object orientation they had subjectively remembered and reproduced on screen themselves from WM. In that sense, our results appear reminiscent of a "generation effect" (for a review, see refs. 14,57, the finding that self-generated information is particularly memorable[58]), which may have been more pronounced after temporary deprioritization. In Experiment 3 (n = 107), we explored whether another common type of visual WM testing (delayed comparison), which does not involve active reproduction of the WM information, may induce subsequent LTM benefits as well.

Except for the difference in WM testing and minor changes to the WM trial timings (see *Methods*), the design of Exp. 3 was identical to Exp. 1. The key difference was that the WM probes in Exp. 3 differed only slightly (+/- 14°) from the original sample orientation, and participants were asked to indicate with a single button press whether the difference was clockwise or counterclockwise (clockwise/counterclockwise; Fig. 4a, right). Thus, whereas the WM probes in Exp. 1 and 2 were uninformative about the original sample orientation, the probes in Exp. 3 did repeat (approximate)

**Fig. 4 | WM and LTM performance in Experiment 3 (n = 107 participants). a** *Left*, WM task performance. Accuracy is shown as percentage correct responses; otherwise same conventions as in Fig. 2a. We observed no significant differences between conditions (see *Results* for details). *Right*, In the WM tests in Exp. 3, participants indicated whether the WM probe orientation was changed (+/-14°) clockwise or counterclockwise relative to the WM sample. (Stimulus material from the same source as in Figs. 1–3; Bank of Standardized Stimuli [BOSS][46]; licensed under CC BY–SA 3.0.) **b** LTM test performance (continuous report), same conventions as in Fig. 2b. While the results showed significant load- and primacy effects, there was no benefit of WM testing (tested vs. not probed) and no effect of WM priority (WM Test 2 vs. 1).

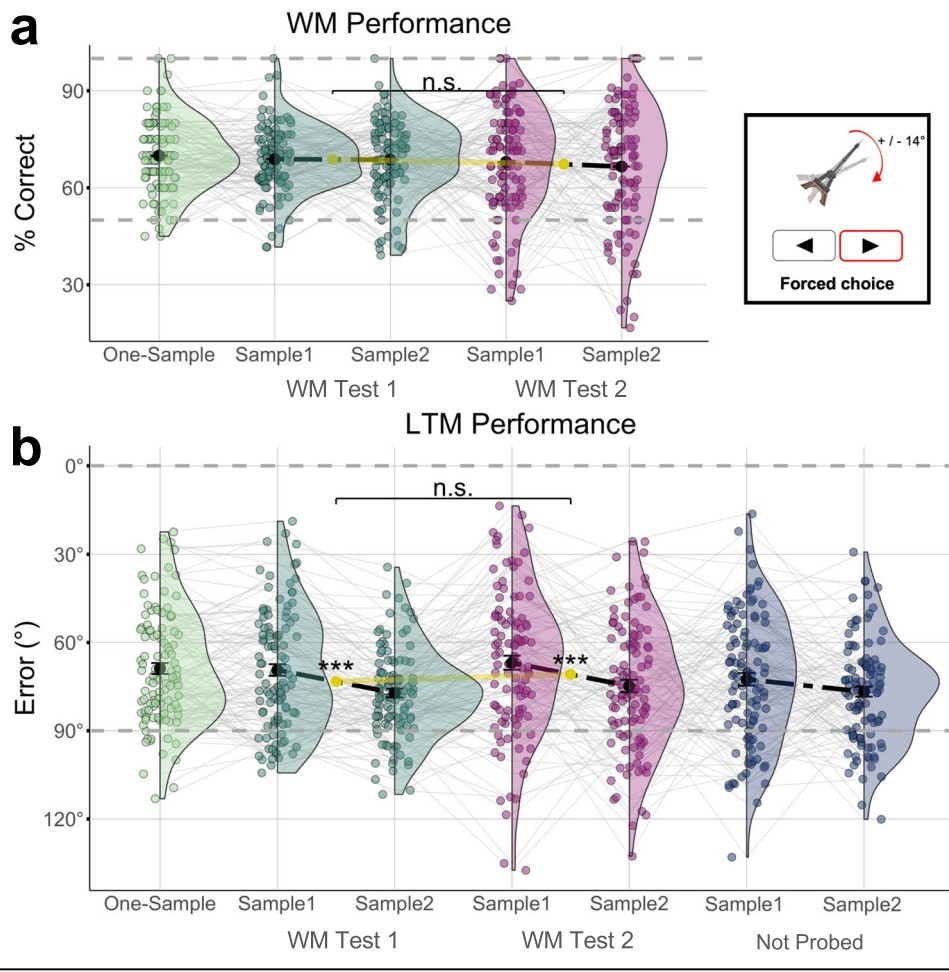

**WM performance.** Unlike in Exp. 1, the WM performance in Exp. 3 was not significantly modulated by load or priority (Fig. 4a). Descriptively, the percentage of correct responses was highest in one-sample trials (M = 70.00%, SE = 1.10%), followed by Test 1 and Test 2 on two-sample trials (M = 69.00%, SE = 0.008 and M = 67.30%, SE = 1.20%), but the differences were not statistically significant [one-sample vs. Test 1: t(106) = 0.901, p = 0.452, 95% CI [-0.01, 0.03], d = 0.087; Test 1 vs Test 2: t(106) = 1.217, p = 0.452, 95% CI [−0.01, 0.04], d = 0.118]. Focusing on the two-sample trials, a 2 × 2 repeated measures ANOVA with the factors WM Sample Position (1/2) and Test (1/2) yielded no significant main effects [Sample Position: F(1,106) = 0.309, p = 0.580, η2 = 0.002, 95% CI [0.00, 1.00]; Test: F(1,106) = 1.164, p = 0.283, η2 = 0.011, 95% CI [0.00, 1.00]] and no interaction [F(1,106) = 0.213, p = 0.645, η2 = 0.002, 95% CI [0.00, 1.00]]. Thus, albeit WM performance in Exp. 3 was significantly above chance [t(106) = 28.771, p < 0.001, 95% CI [0.67, 0.70], d = 2.781], it was hardly modulated by task factors (for similar null-results using a recognition test, see ref. 29).

**LTM performance.** The subsequent LTM test procedure in Exp. 3 was identical to that in Exps. 1 and 2. Compared to Exp. 1, the overall LTM accuracy in Exp. 3 was significantly lower [M = 72.14°, SE = 1.137°, t(236.6) = 10.168, 95% CI [14.44, 21.45], d = −1.242, p < 0.001]. Furthermore, unlike the previous experiments, Exp. 3 showed only a weak WM-testing benefit relative to unprobed items [Fig. 4b; M = 70.95°, SE = 1.88° vs. M = 74.68°, SE = 1.45°; t(106) = −2.00, p = 0.048, 95% CI

[0.03, 7.42], d = −0.193; paired t-test comparing Test 2 vs. unprobed samples, uncorrected]. Comparing the testing effects in Exps. 1 and 3 directly, a mixed-effects ANOVA with the between-subjects factor Experiment (Exp. 1 vs. 3) and the within-subjects factor WM-testing (Tested vs. not probed) showed significant main effects for both factors [Experiment: F(1,292) = 76.202, p < 0.001, η2 = 0.207, 95% CI [0.14, 1.00]; Testing: F(1,292) = 51.062, p < 0.001, η2 = 0.149, 95% CI [0.09, 1.00]] as well as a significant interaction [F(1,292) = 17.755, p < 0.001, η2 = 0.060, 95% CI [0.02, 1.00]], which confirms that the delayed-comparison WM testing in Exp. 3 had less benefits for subsequent LTM than the continuous-report WM tests in Exp. 1.

Across conditions in Exp. 3, participants were slightly more accurate in recalling the orientations from one-sample WM trials (M = 68.83°, SE = 1.87°) compared to two-sample WM trials [Tests 1 and 2 combined, M = 72.34°, SE = 1.54°; t(106) = −2.38, p = 0.019, 95% CI [−6.43, −0.58], d = -0.230; Fig. 4b]. However, focusing on two-sample trials, unlike in Exp.1, we found no significant LTM benefit for samples probed in WM Test 2 (M = 70.95°, SE = 1.88°) compared to WM Test 1 [M = 73.02°, SE = 1.55°; t(106) = −1.492, p = 0.139, 95% CI [−4.81, 0.68], d = −0.144]. A 2 × 2 ANOVA (specified analogously as above) showed a main effect of Sample Position [1/2; F(1,106) = 25.056, p < 0.001, η2 = 0.191, 95% CI [0.09, 1.00]], indicating a primacy effect, but no main effect of WM Test [1/2, F(1,106) = 3.056, p = 0.083, η2 = 0.028, 95% CI [0.00, 1.00]] and no interaction between the two factors [F(1,106) = 0.002, p = 0.969, η2 < 0.001, 95% CI [0.00, 1.00]]. Post-hoc tests showed the primacy effect to be significant both for samples probed first and second in WM [Test 1 and Test 2; t(106) = −4.553, p < 0.001, 95% CI [−11.24, −4.42], d = -0.440 and t(106) = −3.055, p < 0.001, 95% CI [−13.11, −2.79], d = −0.295; Fig. 4b].

Together, while the LTM results of Exp. 3 replicated a primacy effect, the different WM testing procedure eliminated the LTM benefit for deprioritized WM information that we found in the previous experiments. In fact, unlike in Exps. 1 and 2, WM testing in Exp. 3 barely had an LTM benefit at all. These observations corroborate that the LTM benefits for deprioritized WM information in Exp. 1 and 2 were likely mediated by retrieval and/or self-generation processes associated with continuous reporting. At the same time, Exp. 3 showed that the repeated presentation of (approximate) sample information during the WM tests was not sufficient to induce a substantial LTM benefit, for neither the deprioritized nor the deprioritized WM samples.

## Discussion

To summarize our main findings, using a WM-LTM paradigm with oriented objects, we found that although attentional deprioritization reduced immediate WM recall accuracy, it increased subsequent LTM recall performance. This pattern was observed both when WM priority was manipulated through testing order (Exp. 1) or retro-cues (Exp. 2). More specifically, deprioritization appeared to enhance the LTM benefit of WM retrieval, and led to a stronger long-term memory of the information that had been remembered at the WM test (via continuous report). In contrast, no clear LTM enhancement was observed when we tested WM with a simpler (binary choice) delayed comparison task (Exp. 3). In addition, the LTM results in all experiments showed a 'primacy' pattern, such that items that had occurred at the beginning of a WM-trial episode were later recalled better. A similar primacy effect was also evident in WM recall, but only for deprioritized information. Together, our findings highlight various aspects in which WM retrieval of deprioritized information—as opposed to prioritized information—resembles retrieval from episodic LTM.

It is well-established that temporary deprioritization of WM content reduces its accuracy[36,47,59], a finding we also replicated here (Exp. 1 and 2). By intuition, one might assume that if information is deprioritized in WM, it is also less likely to be encoded into a durable long-term memory. Indeed, several recent studies, mostly using recognition tests, found that unprioritized WM contents were later remembered less well than prioritized ones[24–29]. Here, using a more recall-like WM testing format (continuous reports), we found the opposite: deprioritization in WM paradoxically improved subsequent LTM recall. At face value, this counterintuitive result is reminiscent of previous work on the "McCabe effect"[33–35,60], where an intermittent distractor task during word-list learning impaired the words' immediate (WM-like) recall, but improved their later (LTM-like) recall after a longer delay. The McCabe effect has been explained in terms of 'covert' retrieval of the WM information back into the focus of attention[35,61] after it had temporarily been stored in 'activated LTM' (e.g. refs. 16,36). In our present experiments with non-verbal materials, covert retrieval may have contributed to the LTM results in Exp. 1, where the added WM delay (Delay 2) could have provided additional opportunity for such processing. However, as outlined below, the entirety of our results across experiments indicates that LTM benefits for deprioritized WM contents arose from *overt* WM testing, specifically with continuous reports. As such, our results may also help explain occasional failures to find a McCabe effect in some previous studies with nonverbal materials, where WM was probed with old/new recognition only[62].

We found subsequent LTM benefits after WM deprioritization not only when priority was manipulated through WM-testing order (Exp. 1), but also when using retrospective cues in Exp. 2. With the latter experiment design, the effect showed directly as a stronger WM-testing benefit (relative to untested/not probed samples) for deprioritized information. This result can not be easily explained by differences in covert retrieval and/or WM delay length, but can be attributed to the (overt) WM testing proper. In line with this interpretation, in Exp. 1 and 2, participants' LTM reports were more similar to their own previous WM reports than to the original WM sample information, and this 'bias' was increased after WM deprioritization. In other words, after WM deprioritization, the participants appeared to

show a stronger (overt) "generation effect" ([63,64], for review, see ref. 14), which further underscores the role of overt testing/reporting in explaining our results. Notably, although LTM bias towards the (imperfect) WM reports in principle reflects a source of error, the WM errors were small enough (relative to the LTM error) for such bias to still go along with an objective LTM benefit for the deprioritized WM materials. An alternative explanation in terms of canonical orientation biases[54,65–67] was not supported by our present data (see Supplementary Fig. 1 and Supplementary Analysis 1). Finally, we observed no LTM boost for deprioritized WM information —and hardly any WM-testing benefits at all—in Exp. 3, which was near-identical to Exp. 1 but used a different WM testing procedure that relied less on active retrieval and/or self-generation. Together, our results underscore how overt WM testing may affect subsequent LTM, and show that the long-term consequences of WM testing can depend—in seemingly counter-intuitive ways—on the WM information's attentional state.

The present WM-testing effects, particularly for deprioritized information, show notable parallels to classic (LTM-)testing or "retrieval practice" effects in the episodic (long-term) memory literature. LTM-testing effects are known to be more pronounced if successful retrieval practice of the material is more difficult ([68,69], for review, see ref. 12). In a similar vein, the present WM-testing effects were strongest for those materials that were hardest to remember in the WM task (i.e., the deprioritized materials). Further in line with a "retrieval-effort" account, LTM-testing effects are typically larger with recall than with recognition testing[70–72], and we likewise observed greater benefits with a recall-like WM test (Exp. 1 & 2) than with simpler (binary) sample-probe judgments (Exp. 3). Possibly, active recall also involves the generation of effective retrieval cues, resulting in a 'deeper' processing of the WM information[7] which leads to better subsequent memory. Lastly, LTM-testing effects are typically shown relative to a "restudy" baseline where the memory material is presented again without retrieval requirements[12,72]. Our WM experiments did not include dedicated restudy conditions; however, the WM probes in Exp. 3 did reshow the sample information in reasonable approximation (Fig. 4a) to allow for restudying it. The lack of clear testing effects in Exp. 3 thus renders it unlikely that the robust LTM benefits in Exp. 1 & 2 would also have occurred under restudy conditions. Here, in the context of our WM task trials, we cannot rule out that different test (or restudy) formats might also lead to differences in how effortfully participants would encode and/or maintain the WM information. These limitations notwithstanding, the long-term consequences of WM testing in our tasks showed many of the hallmarks of classic (episodic) retrieval-practice and align well with existing accounts of LTM-testing effects (e.g., retrieval effort theories[12]).

Another potential parallel between WM retrieval of deprioritized information and LTM retrieval in our tasks appeared evident in the extent to which the first or the second sample in a WM trial was remembered better. In the final LTM tests, in all our experiments, we observed a clear 'primacy' benefit for the first-presented WM sample. As a possible explanation, the first sample marked the beginning of a new (WM-)task episode, which may have promoted its contextual encoding into episodic LTM. Of note, such within-trial primacy effects were clearly evident also for samples that were not probed in the WM task. This supports a view that the primacy effects reflected episodic/contextual encoding factors[73], unlike the retrieval-induced phenomena discussed in the previous paragraphs. Interestingly, in Exp. 1, a moderate primacy effect was evident also in WM recall, but only after deprioritization (WM Test 2). The WM recall of prioritized information (WM Test 1) in contrast showed, if anything, a (non-significant) recency effect, i.e., better recall of the last-presented sample. A similar but non-significant difference in primacy/recency was also seen in Exp. 2 which had a smaller participant sample. Albeit speculative, these observations may suggest that the role of episodic context factors, in terms of within-trial primacy, increased from prioritized WM over deprioritized WM to LTM recall, which adds to the apparent similarities between the latter two.

There exists a range of views on how unattended WM storage is implemented mechanistically in the brain[37,74–78] and the extent to which

the underlying processes are distinguished (or not) from episodic LTM remains debated ([79,80], for a related proposal, see ref. [39]). A previous study found no evidence that unattended WM maintenance would improve subsequent LTM[29], and we likewise observed no LTM-benefits for deprioritized materials (see Exp. 2, not probed items) unless the material was explicitly tested. We thus found no evidence that unattended WM contents would make stronger contact with LTM through unattended storage per se. Instead, the LTM benefits manifested only when the material was actively recalled from its deprioritized WM state. The observed similarities to LTM retrieval are consistent in principle with a view that the deprioritized WM information may have been maintained in a LTM-like storage state[37,79], where bringing the information back into the focus of attention[16,36] may resemble episodic memory retrieval. Alternatively, our results may indicate that retrieval from dedicated "unattended" WM storage formats (e.g. refs. [74,78]) benefits later LTM recall through yet unknown mechanisms. Specifically, it has been proposed that unattended WM information may undergo representational transformation (e.g.[78,81,82,]) and/or involves "activity-silent" storage in short-term synaptic weight patterns[74,83]. Further work using neural recordings will be needed to differentiate between these possibilities.

### Limitations

Although our study provides insights into how WM retrieval influences LTM retention, several potential limitations should be acknowledged. First, our conclusions are based solely on behavioral data and can therefore only be cautiously interpreted with regard to the ongoing debate on how WM contents are stored during periods of inattention. Future work combining our paradigm with neural measures can be instrumental for clarifying the underlying neural processes. Second, although our experiments used complex real-world images as stimuli, our memory test(s) probed only a single feature (i.e., orientation). It therefore remains an open question whether similar "WM-testing" effects would emerge when memory is probed for other (i.e., perceptual or semantic) stimulus aspects. Finally, in comparing results between experiments, we cannot rule out that different test (or restudy) formats might also lead to differences in how effortfully participants would encode and/or maintain the WM information, which could be mitigated by varying test formats trial-by-trial in future designs.

### Conclusion

To conclude, factors that promote (or hinder) subsequent remembering are of central concern in basic memory research, but also in applied contexts such as the educational sector. Here, we showed that recalling information from WM can promote its long-term retention, particularly if the WM information has temporarily not been in the focus of attention. Beyond resembling classic LTM-"testing" effects, our results join other findings that some memory operations (e.g., 'replay'[84,85]) seem to favor weaker, or more distant memories (see also ref. [86]) despite them potentially being less accurate. An intriguing question for future work is how WM retrieval of deprioritized information intersects with processes thought to underlie long-term memory and learning on the neural level.

### Data availability

The behavioral data supporting the findings of this study are available as .json files for each participant and are available on https://doi.org/10.12751/g-node.3p3ryv.

### Code availability

The analysis code and experiment code are available on GitHub and archived on Zenodo: https://doi.org/10.5281/zenodo.13867138. and https://doi.org/10.5281/zenodo.13867798.

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

## Acknowledgements
We thank Maik Messerschmidt and Philip Jakob for technical assistance with online data collection, Stefan Appelhoff for advice regarding data sharing and Open Science, Marit Petzka for help with statistical review, and Aleksandra Zinoveva for help with stimulus selection and administration. We also thank Or Yizhar, Juan-Linde-Domingo, Maria Wimber, and Lukas Muttenthaler for helpful comments and discussions. This research was supported by European Research Council Consolidator Grant ERC-2020-COG-101000972 (B.S.) and Deutsche Forschungsgemeinschaft (DFG) Grant 462752742 (B.S.). The funders had no role in the study design, data collection and analysis, or decision to publish.

## Author contributions
F.J.B.: conceptualization, data curation, formal analysis, investigation, project administration, validation, visualization, writing—original draft, writing—review and editing. B.S.: conceptualization, funding acquisition, methodology, project administration, resources, supervision, writing—original draft, writing—review and editing.

## Funding

## Competing interests
The authors declare no competing interests.
