## [Transparent Peer Review file · Communications Psychology]

Long-Term Effects of Working Memory Retrieval From Prioritized and Deprioritized States.

Corresponding Author: Ms Frieda Born

Version 0:

Decision Letter:

Dear Ms Born,

Thank you for your patience during the peer-review process. Your manuscript titled "Long-Term Effects of Working Memory Retrieval From Prioritized and Deprioritized States." has now been seen by 3 reviewers, and I include their comments at the end of this message. They find your work of interest but raised some important points. We are interested in the possibility of publishing your study in Communications Psychology, but would like to consider your responses to these concerns and assess a revised manuscript before we make a final decision on publication.

We therefore invite you to revise and resubmit your manuscript, along with a point-by-point response to the reviewers. Please highlight all changes in the manuscript text file.

Editorially, we consider the need for stronger theoretical framing to be central to the revision. Key methodological concerns should also be addressed, including incorporating precision analyses alongside accuracy, using bias and mixture modeling to disentangle sources of recall error, clarifying descriptions of the memory tests, and testing the robustness of a key interaction. While one reviewer suggests additional data collection may be needed to disentangle retrieval demands from encoding/maintenance effort, but we believe this is unnecessary if limitations are acknowledged and interpretations are appropriately cautious. Improving figure clarity, reporting, and data visualization is also essential.

The reviewers comment on the reporting and interpretation of statistics; as you address these issues, please note our policy on statistics reporting and interpretation, which requires full statistics reporting for all statements, and does not permit interpreting null results in NHST as indicative of the absence of an effect or difference. These statements require positive evidence for the null, derived from Bayesian statistics or equivalence tests. NHST that fail to reach conventional levels of statistical significance should of course be reported in full, but may not be interpreted. Detailed guidance is included in the attached checklist and on our website: <https://www.nature.com/commspsychol/submit/submission-guidelines#statistical-guidelines>.

I am attaching an Editorial Requests Table that details critical reporting requirements for the revised manuscript. Please attend to each item and ensure your manuscript is fully compliant. If your revised manuscript is not aligned with these requests on major issues, such as those concerning statistics, it may be returned to you for further revisions without re-review.

Please submit the following items:

- Revised manuscript
- Point-by-point response to the referees' comments

- Cover letter (as a separate document)
- [Nature Research Reporting Summary](https://www.nature.com/documents/nr-reporting-summary.zip)
- [Editorial Policy Checklist](https://www.nature.com/documents/nr-editorial-policy-checklist.pdf)
- Completed Editorial Request Table (attached).

via this link: Link Redacted .

Additional guidance is available in our style and formatting guide [Communications Psychology formatting guide](https://www.nature.com/documents/commpsychol-style-formatting-guide-accept.pdf).

Best regards,

Jesse Rissman

Jesse Rissman, PhD
Editorial Board Member
Communications Psychology
orcid.org/0000-0001-8889-5539

REVIEWER EXPERTISE:

Reviewer #1 working memory
Reviewer #2 working memory / LTM interactions
Reviewer #3 working memory / LTM interactions

REVIEWER REPORTS:

Reviewer #1 (Remarks to the Author):

Review of COMMSPSYCHOL-24-0709-T, "Long-Term Effects of Working Memory Retrieval From Prioritized and Deprioritized States".

The manuscript reports the results of three experiments designed to elucidate the "factors that determine whether information temporarily held in working memory (WM) is transferred to long-term memory (LTM)". Note that this hypothesis presumes the construct of information transfer from WM to LTM, as in "store" based models of memory (Broadbent; Atkinson & Shiffrin; Baddeley & Hitch). Below, alternative views more consistent with state based models of memory (Cowan; Oberauer) are discussed. In the experiments, healthy adults had to remember either one or two visually presented objects and their orientations, and then, after all of the WM trials and 1 minute of mental arithmetic, a surprise subsequent LTM test assessed the long-term retention of participants' memory of the objects and their orientations from the WM test, and whether retention varied if items were held and retrieved in a prioritized or deprioritized state and were or were not tested on the initial WM test. The authors concluded that there was a LTM benefit of testing both for prioritized and deprioritized items on the WM, which was stronger for deprioritized items.

Methodologically, the experiments seemed to be well designed, well powered, and analyzed appropriately. The authors conclusions regarding the data are sensible. Additional considerations (e.g., that the notion of items maintained and retrieved on the WM test trials being differentially re-encoded and transferred to a separate LTM store may not be the most biologically plausible conceptualization) to take into account are discussed below. The manuscript seemed to be pretty clear

and well written. It is also appreciated that aspects of the review of relevant literature is quite scholarly by including some reference to the extensive (and excellent) literature on this topic that was done in the 60s and 70s. Some additional references that may also be of interest to the authors are provided below.

In sum, there is much to like about these experiments, the analyses, results, and interpretations. Some suggestions for revisions which are designed to help clarify and strengthen the impact of this important work are as follows, in no particular order:

1. That subsequent LTM was better for successfully recalled deprioritized items on the WM task is consistent with the LTM account of reactivating deprioritized items from WM (but see LaRocque et al., 2015, *Memory & Cognition*; Chao, Xu, & Rose 2023, *QJEP*; for conflicting results).
2. Some clarification is needed with the description of the memory assessments. call it recognition so people understand that this is what it was. Rather than calling the tests in Exp. 3 “simple WM comparisons against a probe”, why not call them the more common terms of a recognition or change-detection test?
3. The authors stated that “the surprisingly larger LTM benefit for deprioritized WM contents may reflect enhanced encoding of the participants’ own subjective WM report – as opposed to the originally presented sample information – into LTM. Others have suggested that retrieving deprioritized (or uncued/unattended/latent) items (held outside of focal attention) is a more effortful, elaborative form of retrieval practice using episodic LTM processes, which benefits subsequent LTM more than reporting prioritized items directly from focal attention (Loaiza & Lavilla, 2021, *JML*; Rose, Buchsbaum & Craik, 2014, *Memory & Cognition*). Readers may find this nuanced interpretation helpful with interpreting results on this topic. Notably, some researchers (e.g., Craik and other advocates for a processing-based approach to the WM/LTM distinction) reject the notions of separate encoding processes for WM ‘encoding’ vs. LTM ‘encoding’ and a transfer of representations FROM working memory TO long-term memory stores. For example, how biologically plausible is it for a neurocognitive mechanism to exist that makes copies WM representations to transfer it to a separate/distinct LTM store in the brain? In this modern era of network neuroscience, with a better understanding of the nature of distributed representations and processes and how they dynamically change over time from highly activated processed states to deprioritized, latent (“activity-silent”?) states, consideration of this nuanced interpretation is recommended.
4. In addition to the referenced Cowan, Oberauer, and Beukers et al. papers, the recent Foster, Awh & Vogel Oxford Handbook chapter and the Rose 2020 *Current Directions* paper espouse some unique, but also complementary perspectives, regarding the role of episodic LTM processes in WM performance that may be of interest to the authors.
5. The authors are right that “creating similar conditions in a WM-task context is difficult because without the expectation of a WM test, there would be no reason for participants to engage in WM maintenance at all, but participants can and do incidentally encode information, which can support short-term retention and affect subsequent LTM. The Rose & Craik, 2012, *JEP:LMC* paper on this topic may be of interest.
6. The notion that the factors of overt WM retrieval, distraction, and cueing have only received relatively little attention is probably overstated. The following papers seem relevant: Rose, Myerson, Roediger, & Hale, 2010; Rose & Craik, 2012; Rose, Buchsbaum & Craik, 2014; Chao et al., 2023. Note that this is earnestly not simply for gratuitous self-promotion, but is to bring the authors’ attention to these papers in case they find them relevant and helpful.
7. On manuscript p. 3 the authors state that previous studies reported no differential LTM benefit of word-list recall after distraction in a ‘complex span’ WM task (Loaiza et al., 2021; McCabe, 2008). However, these papers did show differences in LTM benefits from maintaining and recalling items on complex vs. span tasks; that is the so-called McCabe effect. Some studies even show differences in LTM benefits for WM items maintained and recalled from different states of accessibility (e.g., items from different serial positions, McCabe, 2008). Some clarification with this review seems to be needed.
8. Note that being able to quantitatively measure the amount of influence on subsequent LTM from individuals’ subjective reports on the WM test is a novel contribution of these experiments and should be viewed as a strength of the current research.
9. Another strength is that the experimental paradigms come closer to matching the nature of the memory tests between the WM and LTM tests. Some persnickety reviewers of manuscripts reporting WM and subsequent LTM data in the past have argued that differences between WM and subsequent LTM may have nothing to do with differences in the neurocognitive processes that underlie performance on WM and LTM tests -- the differences may only be due to the fact that WM tasks typically test for recall whereas subsequent LTM tests typically test for recognition (in order to avoid floor effects). While differences between the format of the tests contribute the differences observed, they cannot explain all of the differences between memory for items retrieved on WM tests and on subsequent LTM tests. The current work nicely supports the latter conclusion, especially Exp. 3. For example, in Hartshorne and Makovski’s particularly critical review of this literature, they concluded that differences in subsequent LTM between items processed differently on initial WM tasks is simply due to noise, not any difference in the neurocognitive processes involved in the different situations. Exp. 3 clearly shows that part of the variability in prior studies is likely due to differences in the type of tests (recall vs. recognition).
10. The authors state that their results showed clear subsequent LTM benefits of WM testing with continuous reports and further revealed that participants’ LTM reporting was biased towards their WM reports. But did they also do the bias analyses to see how much responses were biased to the stimulus orientation? The LTM analyses conditionalized on items initially recalled with high vs. low error is helpful. But many studies have shown that there are multiple sources of both attractive and repulsive biases that influence recall error, from the initially encoded stimulus, the recalled response, other items on the trial, items from previous trials, and even the proximity to the cardinal axes. Without redoing the bias analyses to gauge the amount of bias from all of these factors, it is tenuous to strongly conclude that the bias was solely (or primarily) due to participants’ memory of their initial WM response. At the very least, the reader should get more of a review and reference to these issues from the serial dependence and related literatures so that they can find and be made aware of these multiple sources of bias.
11. With regards to the results, the degree of error on the WM and especially the LTM tests seems quite large. Readers may wonder how much guessing is contributing to these distributions and if mixture modeling to separate out guesses from

responses toward the target (precision) or nontarget (swap errors) may reveal clearer and potentially more interesting differences between the sources of errors for items from the different WM conditions. For example, to what extent might the LTM responses be strongly biased towards the closest cardinal axis, as if the orientation of the objects are encoded more categorically (left or right, top or bottom) than precisely in the full 360 degree feature space. The influence of biases to the cardinal axes seems to have a strong influence on WM that differs for items held in a prioritized or deprioritized state (Rhilinger, Xu, Rose, 2023, APP; Bae, 2024).

12. It should be noted that, because the LTM test, presents each object from the WM test and asks participants to recall its orientation, this LTM test is really more of a cued associative recall test than a true free recall test. This should probably be clarified for the readers in the intro, results, or discussion so that they don't miss this important detail in the methods.

13. The "pruning" control analyses in which items were matched on WM performance to equate the degree of error between the conditions and conditionalizing LTM recall on initial recall performance is a nice control analysis. Conditionalizing subsequent LTM recall on WM accuracy has precedence in this literature and should be seen as a strength of the manuscript. Readers and reviewers don't always understand the importance of such control analyses, so the hope of this reviewer is that other reviewers or editors will not make the authors remove these control analyses from the manuscript. The results strengthen confidence in the results.

14. The control analysis on p. 11 lines 372-377 seems unnecessary. It would be better and more interesting to assess the degree of bias in recall from the other item's orientation on that trial. These results could help inform the somewhat mixed literature on serial dependence effects on WM and LTM (Bae & Luck, 2017, APP, 2019 Psych Science; Rhilinger, Xu, & Rose, 2023, APP; Bae, 2024, APP).

15. Finally, the authors appeal to the notion of "retrieval effort" as a critical process that contributes to the benefit to subsequent LTM. One suggestion is that, rather than just ascribing the benefit to effort, an alternative mechanism is the notion of a more elaborative form of retrieval being involved in conducting a search of memory to recover deprioritized items by self-generating retrieval cues (as the authors suggest with the reference to the generation effect). The critical distinction is that having to search memory and generate effective retrieval cues, which is more likely to involve a deeper, conceptual level of processing than simply reporting a prioritized item that was continuously maintained in a high level of activation (in the "focus of attention"), which is more likely to involve a shallower, more perceptual than conceptual level of processing, e.g., Craik & Jacoby's 1975 process view of short-term retention: [https://www.larryjacobson.ca/images/craik %26 jacobson \(1975\).pdf](https://www.larryjacobson.ca/images/craik%26jacobson(1975).pdf)

I hope these comments are found to be helpful with revising this manuscript and the authors' future work in this important area.

Best,
Nathan S. Rose
U. Notre Dame

Reviewer #2 (Remarks to the Author):

The authors aimed to demonstrate how the prioritization of contents in working memory (WM) impacts long-term memory (LTM) retention. In the visual WM task, one or two items (i.e., one- or two-sample conditions) were encoded one-by-one and maintained in WM during a delay period until the angle of the items was tested at the end of each trial with self-generated angle responses using continuous measures (Exp. 1, 2) or 2-choice response (Exp. 3). The same items were then tested for their presented angles in the subsequent LTM test with the same continuous measures. The prioritization in WM was defined in the 2-sample condition based on either WM testing order or retro-cue, in which first tested or cued items became prioritized, and thus the remaining items were deprioritized, and a subset of trials was not tested in the WM probe task to serve as a baseline. The main findings showed that attentional deprioritization reduced immediate WM recall accuracy, while it increased subsequent LTM recall performance.

While the objective of the study - understanding how different prioritization states of contents in WM impact their LTM retentions - is important, and the results are interesting, I still have a few concerns and suggestions.

Major points

1. My major concern is that the framework is not clear enough to understand either the objectives or the hypotheses of the study. Although the authors introduced the McCabe effect as a framework—explaining how less accessible WM contents due to distractors were better retrieved in LTM—there are still gaps in the establishment of hypotheses for visual WM prioritization.

This lack of a clear framework also makes it difficult to identify the study's objectives. Clarifying these objectives could help us better understand how this research contributes to further insights into WM prioritization effects on LTM.

For example, WM deprioritization has been studied within the "activity-silence state" framework, demonstrated using similar WM task paradigms (Lorenz et al., 2021; Mallett & Lewis-Peacock, 2019; Rose et al., 2016; Wan et al., 2024). The framework posits that deprioritized WM representations enter a silence state in which they are deactivated but still accessible in WM. Such a state makes these representations less vulnerable to memory updates, which could result from neural interference caused by perceptual inputs or memory competitions, and ultimately improves LTM compared to prioritized WM contents.

More recently, the "representational transformation" account has been proposed, supported by neural and simulation evidence. This account suggests that deprioritized representations are transformed into orthogonal representational

structures or projected into a different neural space (Piwek et al., 2023; Wan et al., 2022; Yu & Postle, 2021). The representational transformation account is further connected to the primacy effect, in which the initially encoded “sample 1” might be represented in a neural space parallel to the LTM representational space, while “sample 2” might occupy a neural space orthogonal to the original. Hypothetically, the initially encoded representation is supported by original representational structures in LTM, though this has only been demonstrated through model simulations.

Additionally, the McCabe effect appears fundamentally different from the WM deprioritization effect at face glance. The McCabe effect is explained by repeated reactivation of WM contents (i.e., covert retrieval) as a mechanism to protect WM targets against distractors, whereas deprioritization promotes WM deactivation. The potential underlying mechanisms that WM deprioritization might share with the McCabe effect should be clearly introduced.

2. In the introduction, the main factors—encoding primacy and WM state priority—were not clearly defined, and the predicted impact on WM-LTM relationships was insufficient. More specifically, it remains unclear why encoding order was manipulated or how this factor would influence priority-based memory updates. If WM priority is sole focus of the research, simultaneous encoding of two items would provide a more straightforward approach. Additionally, this method could better equalize WM maintenance durations for both items compared to Exp. 2, where the first encoded item was maintained longer in WM, potentially still introducing a confound.

3. Although the authors used continuous measurements for memory performance, the data was only analyzed for accuracy. Continuous response data can provide both accuracy and precision metrics, offering a more comprehensive understanding of memory performance. For example, prioritized WM items might exhibit greater accuracy but reduced precision, while deprioritized items could show the opposite pattern. Incorporating precision as a predictor of LTM effects could significantly enhance the study’s findings.

4. In Exp. 1, it is worth considering whether the worse LTM performance for the first tested items (WM Test-1, prioritized items) could be attributed to retroactive interference from Test-2 item retrieval rather than the WM prioritization factor. To control this possible confound, the LTM data for WM Test-1 trials in the two-sample condition could be analyzed based on the presence or absence of the second WM probe test. Furthermore, based on the retroactive interference account, it is possible that LTM for Test-1 items was biased toward Test-2 items when WM Test-2 was present (Zhang & Lewis-Peacock, 2022). If such interference exists, it should be appropriately controlled, for example, by covarying it out in the analysis.

5. While I appreciate the innovative WM task paradigm, it remains unclear how these modified components contribute to a novel or deeper understanding of WM prioritization.

Minor points

1. There are several suggestions regarding the figures.

a. The arrangement of the plots emphasizes the primacy effects by highlighting the differences between sample 1 and sample 2. For Fig. 2, it would be easier to interpret the results if the two-sample data were grouped by encoding order (i.e., sample 1 vs. sample 2) first, with each group including WM Test 1, WM Test 2, and Not Probed (NP) conditions. This arrangement would also allow for clearer comparisons against the NP baseline trials. A similar arrangement could be applied to Fig. 3, with the data grouped by encoding order (sample 1 vs. sample 2), WM probe status (probed vs. not probed), and retrocue status (cued vs. uncued) in sequence. This structure would better emphasize WM prioritization differences, particularly the contrast between cued and uncued conditions.

b. The current color coding for the plots is not very effective. I recommend using warm colors for the prioritized condition and cool colors for the deprioritized condition. Reduced saturations or transparency within the same color scheme could be applied to represent the not-probed conditions. Although higher transparency was applied for LTM compared to WM results, I do not think using the same colors for WM and LTM leads to confusion. Instead, transparency variations could be used to distinguish between WM probed and not-probed results. Additionally, applying the same color scheme to the WM task diagram in Fig. 1, along with a legend indicating the colors for prioritized and deprioritized conditions, would make the prioritization manipulation easier to understand.

c. The upper bars across conditions for the significance notations are unclear, as it is not evident whether they represent interactions or main effects.

d. In Fig. 2c, the plot showing the LTM-recall bias toward the WM report (i.e., similarity) is difficult to interpret. It would be clearer if the distributions of the differences between the memory report (i.e., either WM or LTM) and the original samples (adjusted to a 0-point baseline), were plotted in separate sub-figures. This would allow for a better understanding of whether the LTM recall was biased toward the WM report or the original samples.

2. There is a typo on page 14 (464): Fig. 1c seems to refer to Fig. 3c instead.

References

- Lorenc, E. S., Mallett, R., & Lewis-Peacock, J. A. (2021). Distraction in Visual Working Memory: Resistance is Not Futile. *Trends in Cognitive Sciences*, 25(3), 228. <https://doi.org/10.1016/J.TICS.2020.12.004>
- Mallett, R., & Lewis-Peacock, J. A. (2019). Working memory prioritization impacts neural recovery from distraction. *Cortex*, 121, 225–238. <https://doi.org/10.1016/J.CORTEX.2019.08.019>
- Piwek, E. P., Stokes, M. G., & Summerfield, C. (2023). A recurrent neural network model of prefrontal brain activity during a working memory task. *PLOS Computational Biology*, 19(10), e1011555. <https://doi.org/10.1371/JOURNAL.PCBI.1011555>
- Rose, N. S., LaRocque, J. J., Riggall, A. C., Gosses, O., Starrett, M. J., Meyering, E. E., & Postle, B. R. (2016).

Reactivation of latent working memories with transcranial magnetic stimulation. *Science* (New York, N.Y.), 354(6316), 1136–1139. <https://doi.org/10.1126/SCIENCE.AAH7011>

Wan, Q., Ardan, A., Fulvio, J. M., & Postle, B. R. (2024). Representing Context and Priority in Working Memory. *Journal of Cognitive Neuroscience*, 36(7), 1374–1394. https://doi.org/10.1162/JOCN_A_02166

Wan, Q., Menendez, J. A., & Postle, B. R. (2022). Priority-based transformations of stimulus representation in visual working memory. *PLOS Computational Biology*, 18(6), e1009062. <https://doi.org/10.1371/JOURNAL.PCBI.1009062>

Yu, Q., & Postle, B. R. (2021). The Neural Codes Underlying Internally Generated Representations in Visual Working Memory. *Journal of Cognitive Neuroscience*, 33(6), 1142. https://doi.org/10.1162/JOCN_A_01702

Reviewer #3 (Remarks to the Author):

Summary:

This manuscript comprises three experiments designed to test the impact of deprioritization and retrieval effort in working memory (WM) on later retrieval from episodic long-term memory (LTM). Specifically, participants completed an online study where trials of one or two objects presented in different orientations were followed by immediate WM tests and a surprise LTM test. The results of Experiment 1 showed that the error in reproducing the orientation of the objects from WM was unsurprisingly greater when participants were tested on two objects than one object, but the reverse was true in LTM: Reproduction error was reduced for objects that were tested second (and thus temporarily deprioritized) versus first. As there was also a primacy effect, Experiment 2 used retro-cues to attempt to disentangle the impact of prioritization and position in the trial, which indeed showed a greater LTM benefit of reduced reproduction error for uncued (i.e., deprioritized) objects versus cued objects. Finally, Experiment 3 showed that these effects are subject to the nature of the WM test, such that the key results of Experiment 1 were nullified when using a simple recognition test instead of reproduction. The authors interpret the findings as consistent with a retrieval effort explanation of the importance of WM for LTM.

Evaluation and major comments:

This was an interesting set of experiments that was straightforward to follow and well situated in the literature on this important WM/LTM topic. Most of my constructive comments further below are points of clarification, but the potentially trickier ones regard the key results on which the major theoretical interpretations hinge.

There are a few points on this issue of interpretation. First, it made sense to use retro-cues in Experiment 2 to follow up Experiment 1, but the key interaction between WM testing and cueing (lines 477-479) is an ordinal one, which are known to be difficult to unambiguously interpret (Loftus, 1978; Wagenmakers et al., 2012). In other words, it is possible this interaction may be removable when transforming the data to another scale that does not have the same potential psychological ceiling/floor as reproduction error. Indeed, key interactions in this WM/LTM topic have been transformed away in previous work (e.g., Labaronne et al., 2023), suggesting that one should be cautious to interpret interactions that may be exacerbated by mere issues of scale. So, my advice here would be to transform the data to another scale to ensure that this key interaction still occurs and is not removable.

The second issue is that quite a lot of heavy lifting in the interpretations rests on null p-values, which also cannot be taken as evidence for the null. Is there any indication of how strong the evidence is for null in these situations, e.g., by using a Bayesian analytic approach? For example, the fact that the key results of Experiment 1 are nullified in Experiment 3 is interesting but could be an issue of power rather than a true null effect (and I have another question about power further on).

Finally, related to this specific finding of nullified results in Experiment 3, I think that the conclusion that overt retrieval effort is interesting but premature when considering the design. Specifically, the method of WM testing is not the only thing that has changed about the study, but perhaps also the participants' approach to the task. If I am asked to make a simple recognition choice based on the content of my WM, I am probably less likely to try to actively encode and maintain as much detail about the orientation of the stimuli compared to a reproduction test as in Experiment 3. Thus, it is currently impossible to know if the null results occurred due to the obvious difference in retrieval demands (as the authors propose) or whether the participants were less proactive in their active maintenance of such detailed information that only became relevant much later in the surprise LTM test. The issue of task engagement has been recently identified as a potentially important moderating factor of the McCabe effect (Cotton et al., 2024), and I myself have tried to disentangle retrieval demands versus task approach in Experiment 3 of the paper cited in this manuscript (Loaiza et al., 2021). It is not immediately obvious to me how to address this issue without conducting another experiment that clearly disentangles encoding/maintenance effort versus overt retrieval effort.

Signed,

Vanessa Loaiza

(Please note that I only sign reviews when the work is obviously tied to my own)

More minor points of clarification:

- Line 128: Could you clarify what you mean by “negative results”?

- Lines 136-139: I do not understand this sentence for two reasons. First, I do not understand what is meant by “no differential LTM benefit” (perhaps different benefit depending on retrieval method?). Second, the second part of the sentence does not seem to follow the first – why is “maintain[ing] visual information” inconsistent or “unlike” these previous studies’ lack of “differential LTM benefit”?

- Lines 139-141: Please provide citations for the “previous studies” in the first part of this sentence.
- Lines 144-148: I did not understand this sentence until later in the paper. I would recommend giving a bit more methodological information earlier on or adjusting it some other way so that the reader does not have to read ahead to understand it.
- Could you clarify the rationale for the sample sizes of the experiments? They seem quite high compared to previous research, which is not a bad thing, but if some a priori power analysis was conducted to justify greater sample sizes that would be good to know. The sample size is also less for Experiment 2 compared to the other experiments, so it would be good to know why this was the case.
- Figure 3d: I assume that these results are collapsed across sample position, but it would be good to break it down by position just to visually verify that effect is indeed stronger for uncued items for both primacy and recency positions.
- Please report effect sizes alongside the simple effects/t-tests (e.g., Cohen’s d).
- Lines 639-642: Harkening back to the previous major issue on interpretation, it would be good to spell out exactly why the effect is better explained by overt versus covert retrieval before jumping to the next point.
- Lines 885-900: I found the explanation of the pruning method vague. I would recommend giving more detail, and ideally also include your analysis script along with your data so that other researchers can reproduce the method in their own work.

References

- Cotton, K., Sandry, J., & Ricker, T. J. (2024). Secondary task engagement drives the McCabe effect in long-term memory. *Memory & Cognition*, 52(8), 1762–1774. <https://doi.org/10.3758/s13421-023-01450-2>
- Labaronne, M., Jarjat, G., & Plancher, G. (2023). Attentional Refreshing in the Absence of Long-Term Memory Content: Role of Short-Term and Long-Term Consolidation. *Journal of Cognition*, 6(1), 5. <https://doi.org/10.5334/joc.246>
- Loaiza, V. M., Doherty, C., & Howlett, P. (2021). The long-term consequences of retrieval demands during working memory. *Memory & Cognition*, 49(1), 112–126. <https://doi.org/10.3758/s13421-020-01079-5>
- Loftus, G. R. (1978). On the interpretation of interactions. *Memory & Cognition*, 6, 312–319. <https://doi.org/10.3758/BF03197461>
- Wagenmakers, E.-J., Kryptos, A.-M., Criss, A. H., & Iverson, G. (2012). On the interpretation of removable interactions: A survey of the field 33 years after Loftus. *Memory & Cognition*, 40(2), 145–160. <https://doi.org/10.3758/s13421-011-0158-0>

Communications Psychology is committed to improving transparency in authorship. As part of our efforts in this direction, we are now requesting that all authors identified as ‘corresponding author’ create and link their Open Researcher and Contributor Identifier (ORCID) with their account on the Manuscript Tracking System prior to acceptance. ORCID helps the scientific community achieve unambiguous attribution of all scholarly contributions. You can create and link your ORCID from the home page of the Manuscript Tracking System by clicking on ‘Modify my Springer Nature account’ and following the instructions in the link below. Please also inform all co-authors that they can add their ORCIDs to their accounts and that they must do so prior to acceptance.

Version 1:

Decision Letter:

Dear Ms Born,

Your manuscript titled "Long-Term Effects of Working Memory Retrieval From Prioritized and Deprioritized States." has now been seen by our reviewers, whose comments appear below. In light of their advice I am delighted to say that we are happy, in principle, to publish a suitably revised version in Communications Psychology.

We therefore invite you to revise your paper one last time to address the remaining concerns of our reviewers and a list of editorial requests. At the same time we ask that you edit your manuscript to comply with our format requirements and to

maximise the accessibility and therefore the impact of your work.

EDITORIAL REQUESTS:

SUBMISSION INFORMATION:

OPEN ACCESS:

* DATA AVAILABILITY:

Link Redacted

Best regards,

Troy Lui, on behalf of

Jesse Rissman

Troy Lui, PhD
Associate Editor
Communications Psychology

Jesse Rissman, PhD
Editorial Board Member
Communications Psychology
orcid.org/0000-0001-8889-5539

REVIEWERS' COMMENTS:

Reviewer #1 (Remarks to the Author):

The authors have generally done a nice job revising their manuscript. Only a couple of minor issues remain.

They state (p. 2, starting line 98) "While many of these studies found that prioritization IMPROVED subsequent LTM (Fan & Turk-Browne, 2013; Jeanneret et al., 2023; LaRocque et al., 2015; Reaves et al., 2016; Strunk et al., 2019; Wang & Van Ede, 2024), others found no such effect (Bartsch et al., 2018; Mao Chao et al., 2023) or even found superior LTM when attention was diverted from the WM information (Rose et al., 2014)." However, as they correctly state later on (p. 25), "A previous study found no evidence that unattended WM maintenance would improve subsequent LTM (LaRocque et al., 2015)..."

The first sentence should move the LaRocque et al., 2015 reference to the Bartsch et al., 2018 and Chao et al., 2023 references. Also note that the Mao is Chao's first name. The Reference is Chao, Xu, Loaiza, & Rose.

This is a nice contribution to the literature. I look forward to seeing forthcoming work on these interesting and important topics.

Reviewer #3 (Remarks to the Author):

Thanks to the authors for addressing my comments. I have read through the revised manuscript, and I think that it satisfactorily addresses these issues that I and the other reviewers raised. I have no further comments, and I recommend publication.

Signed,

Vanessa Loaiza

(Please note that I only sign reviews when the work is obviously tied to my own)

Response to reviews (RIDER R1)

Reviewer 1:

The manuscript reports the results of three experiments designed to elucidate the “factors that determine whether information temporarily held in working memory (WM) is transferred to long-term memory (LTM)”. Note that this hypothesis presumes the construct of information transfer from WM to LTM, as in “store” based models of memory (Broadbent; Atkinson & Shiffrin; Baddeley & Hitch). Below, alternative views more consistent with state based models of memory (Cowan; Oberauer) are discussed. In the experiments, healthy adults had to remember either one or two visually presented objects and their orientations, and then, after all of the WM trials and 1 minute of mental arithmetic, a surprise subsequent LTM test assessed the long-term retention of participants’ memory of the objects and their orientations from the WM test, and whether retention varied if items were held and retrieved in a prioritized or deprioritized state and were or were not tested on the initial WM test. The authors concluded that there was a LTM benefit of testing both for prioritized and deprioritized items on the WM, which was stronger for deprioritized items.

Methodologically, the experiments seemed to be well designed, well powered, and analyzed appropriately. The authors conclusions regarding the data are sensible. Additional considerations (e.g., that the notion of items maintained and retrieved on the WM test trials being differentially re-encoded and transferred to a separate LTM store may not be the most biologically plausible conceptualization) to take into account are discussed below. The manuscript seemed to be pretty clear and well written. It is also appreciated that aspects of the review of relevant literature is quite scholarly by including some reference to the extensive (and excellent) literature on this topic that was done in the 60s and 70s. Some additional references that may also be of interest to the authors are provided below.

In sum, there is much to like about these experiments, the analyses, results, and interpretations. Some suggestions for revisions which are designed to help clarify and strengthen the impact of this important work are as follows, in no particular order:

We thank the reviewer for the positive evaluation of our work and for their helpful suggestions, to which we respond in detail below.

- 1. That subsequent LTM was better for successfully recalled deprioritized items on the WM task is consistent with the LTM account of reactivating deprioritized items from WM (but see LaRocque et al., 2015, Memory & Cognition; Chao, Xu, & Rose 2023, QJEP; for conflicting results).*

We agree and hope to provide a balanced interpretation of our findings along these lines in the revised discussion.

2. *Some clarification is needed with the description of the memory assessments. call it recognition so people understand that this is what it was. Rather than calling the tests in Exp. 3 “simple WM comparisons against a probe”, why not call them the more common terms of a recognition or change-detection test?*

We appreciate this comment; however, we find it important to note that our testing procedure in Exp. 3 was technically not a recognition test (old/new), but required participants to decide whether the orientation of the probe was clockwise or counterclockwise (cw/ccw) compared to the sample (see description on p. 20, Figure 4a *right*, or the methods section of Exp. 3 on page 8). While we agree that this test format shares similarities with recognition testing (as opposed to continuous reports), we think that our labeling as a “comparison” is technically more accurate. We acknowledge the similarity between our testing procedure in Exp. 3 and previous work using recognition tests (La Rocque et al., 2015; Chao et al, 2023) in the discussion.

3. *The authors stated that “the surprisingly larger LTM benefit for deprioritized WM contents may reflect enhanced encoding of the participants’ own subjective WM report – as opposed to the originally presented sample information – into LTM. Others have suggested that retrieving deprioritized (or uncued/unattended/latent) items (held outside of focal attention) is a more effortful, elaborative form of retrieval practice using episodic LTM processes, which benefits subsequent LTM more than reporting prioritized items directly from focal attention (Loaiza & Lavilla, 2021, JML; Rose, Buchsbaum & Craik, 2014, Memory & Cognition). Readers may find this nuanced interpretation helpful with interpreting results on this topic. Notably, some researchers (e.g., Craik and other advocates for a processing-based approach to the WM/LTM distinction) reject the notions of separate encoding processes for WM ‘encoding’ vs. LTM ‘encoding’ and a transfer of representations FROM working memory TO long-term memory stores. For example, how biologically plausible is it for a neurocognitive mechanism to exist that makes copies WM representations to transfer it to a separate/distinct LTM store in the brain? In this modern era of network neuroscience, with a better understanding of the nature of distributed representations and processes and how they dynamically change over time from highly activated processed states to deprioritized, latent (“activity-silent”?) states, consideration of this nuanced interpretation is recommended.*

We are grateful for this comment. In fact, we did not intend to argue for a dual-store account of WM and LTM, and we realize in hindsight that some of our phrasings in the previous manuscript (e.g., “transfer”) may have inadvertently conveyed this impression. We revised the text to remove such potentially misleading phrases for more clarity and neutrality in this respect, and extended our referral to contemporary neurocognitive accounts (such as activity-silent WM, see also our replies to Reviewer 2) in the Discussion. We thank the reviewer for this helpful comment.

4. *In addition to the referenced Cowan, Oberauer, and Beukers et al. papers, the recent Foster, Awh & Vogel Oxford Handbook chapter and the Rose 2020 Current Directions paper espouse some unique, but also complementary perspectives, regarding the role of episodic LTM processes in WM performance that may be of interest to the authors.*

We thank the reviewer for these excellent resources and included them in the Introduction and Discussion sections on page 3 and 25 in the revised manuscript.

5. *The authors are right that “creating similar conditions in a WM-task context is difficult because without the expectation of a WM test, there would be no reason for participants to engage in WM maintenance at all, but participants can and do incidentally encode information, which can support short-term retention and affect subsequent LTM. The Rose & Craik, 2012, JEP:LMC paper on this topic may be of interest.*

We appreciate this comment and agree with the principled possibility of incidental encoding. However, in the context of our paper, we wish to explicitly refer to active maintenance in WM, which is typically assumed to be purposefully goal-oriented. Whether and how testing and/or restudy at short time scales may also affect incidentally encoded information is no doubt an interesting question, but we believe outside the scope of our current study. The suggested reference to Rose & Craik (2012) is included in our revised introduction (p.3).

The notion that the factors of overt WM retrieval, distraction, and cueing have only received relatively little attention is probably overstated. The following papers seem relevant: Rose, Myerson, Roediger, & Hale, 2010; Rose & Craik, 2012; Rose, Buchsbaum & Craik, 2014; Chao et al., 2023. Note that this is earnestly not simply for gratuitous self-promotion, but is to bring the authors' attention to these papers in case they find them relevant and helpful.

Thank you for this comment. To clarify, we did not mean to imply that each of these factors in isolation (overt retrieval, distraction, cueing) had received little attention, but a specific interaction between them, namely, whether the subsequent LTM benefit of overt testing (i.e., the “WM-testing effect”) is modulated by cueing or distraction. We revised the phrasings in the manuscript to clarify this point, and we cite most of the suggested references in the revised introduction and discussion sections. The revised phrasing reads:

“However, compared to the classic testing effects in the LTM literature, the long-term consequences of overt WM testing have thus far received relatively less attention (but see Tozios & Fukuda, 2024; Xie & Reuter-Lorenz, 2024).”

6. *On manuscript p. 3 the authors state that previous studies reported no differential LTM benefit of word-list recall after distraction in a ‘complex span’ WM task (Loaiza et al., 2021; McCabe, 2008). However, these papers did show differences in LTM benefits from maintaining and recalling items on complex vs. span tasks; that is the so-called McCabe effect. Some studies even show differences in LTM benefits for WM items maintained and recalled from different states of accessibility (e.g., items from different serial positions, McCabe, 2008). Some clarification with this review seems to be needed.*

We agree in hindsight that our phrasing of this sentence may have been slightly confusing. What we meant to highlight here was that these previous studies observed an McCabe effect regardless of whether WM was overtly tested or not. We revised the sentence to be clearer; it now reads:

“On the other hand, the abovementioned ‘McCabe effect’ on subsequent LTM has in a few studies also been observed in trials where overt WM testing was omitted (Loaiza et al., 2021, McCabe 2008).”

7. *Note that being able to quantitatively measure the amount of influence on subsequent LTM from individuals' subjective reports on the WM test is a novel contribution of these experiments and should be viewed as a strength of the current research.*

We are glad about the reviewer's highlight of the strengths and novelty of our study – thank you very much.

8. *Another strength is that the experimental paradigms come closer to matching the nature of the memory tests between the WM and LTM tests. Some persnickety reviewers of manuscripts reporting WM and subsequent LTM data in the past have argued that differences between WM and subsequent LTM may have nothing to do with differences in the neurocognitive processes that underlie performance on WM and LTM tests -- the differences may only be due to the fact that WM tasks typically test for recall whereas subsequent LTM tests typically test for recognition (in order to avoid floor effects). While differences between the format of the tests contribute the differences observed, they cannot explain all of the differences between memory for items retrieved on WM tests and on subsequent LTM tests. The current work nicely supports the latter conclusion, especially Exp. 3. For example, in Hartshorne and Makovski's particularly critical review of this literature, they concluded that differences in subsequent LTM between items processed differently on initial WM tasks is simply due to noise, not any difference in the neurocognitive processes involved in the different situations. Exp. 3 clearly shows that part of the variability in prior studies is likely due to differences in the type of tests (recall vs. recognition).*

We wholeheartedly agree – thank you very much for these comments.

9. *The authors state that their results showed clear subsequent LTM benefits of WM testing with continuous reports and further revealed that participants' LTM reporting was biased towards their WM reports. But did they also do the bias analyses to see how much responses were biased to the stimulus orientation? The LTM analyses conditionalized on items initially recalled with high vs. low error is helpful. But many studies have shown that there are multiple sources of both attractive and repulsive biases that influence recall error, from the initially encoded stimulus, the recalled response, other items on the trial, items from previous trials, and even the proximity to the cardinal axes. Without redoing the bias analyses to gauge the amount of bias from all of these factors, it is tenuous to strongly conclude that the bias was solely (or primarily) due to participants' memory of their initial WM response. At the very least, the reader should get more of a review and reference to these issues from the serial dependence and related literatures so that they can find and be made aware of these multiple sources of bias.*

We wish to emphasize that our manuscript indeed does report a detailed analysis of cardinal bias in the Supplemental Material, which may have been overlooked by the reviewer. We focused on cardinal bias because we identified it as a potential source of variance in previous work with similar stimuli and tasks (Linde-Domingo & Spitzer, 2023, Nature Human Behaviour). The analysis showed no evidence that cardinal biases could explain the WM-LTM results pattern in our present study (see Supplementary Figure 1).

In response to the reviewer's suggestion, we also examined whether the WM responses in our task might have been biased toward the orientation of the other, non-tested item (see newly included Supplementary Analysis 1). We found, if anything, evidence for a weak repulsive effect, in that participants tended to report the stimuli to be slightly more distinct from each other than they were. Importantly, however, we found no difference in this pattern between prioritized (Test 1) and deprioritized samples (Test 2), ruling out that it could have explained the LTM bias to the WM report we found, which was notably stronger for Test 2

samples (Fig. 2c). We included this additional observation, which we think strengthens our previous interpretation, in the revised Supplementary Material (Supplementary Analysis 1).

Together, we found no evidence that participants' LTM-bias towards their WM reports (Fig. 2c) would have been driven by generic cardinal biases or inter-item repulsions.

10. *With regards to the results, the degree of error on the WM and especially the LTM tests seems quite large. Readers may wonder how much guessing is contributing to these distributions and if mixture modeling to separate out guesses from responses toward the target (precision) or nontarget (swap errors) may reveal clearer and potentially more interesting differences between the sources of errors for items from the different WM conditions. For example, to what extent might the LTM responses be strongly biased towards the closest cardinal axis, as if the orientation of the objects are encoded more categorically (left or right, top or bottom) than precisely in the full 360 degree feature space. The influence of biases to the cardinal axes seems to have a strong influence on WM that differs for items held in a prioritized or deprioritized state (Rhilinger, Xu, Rose, 2023, APP; Bae, 2024).*

As regards swap errors, we refer to our inspection of bias to the respective other WM sample in the trial (see our response to the reviewer's previous point 9 above), which showed no evidence for systematic misremembering of the other item – to the contrary, we found evidence for a weak repulsion of the two.

As regards bias to the nearest cardinal axes, we wish to note again that our analysis does include a detailed analysis of cardinal bias (see Supplementary Figure 1) for both the WM and LTM tests, which showed no systematic effects.

Finally, the reviewer's comment about guesses encouraged us to inspect response error distributions (both in WM and LTM) to eyeball whether mixture modeling would be expected to reveal additional insights. The error histograms (see below and Supplementary Figure 2 in manuscript) show the WM and LTM reporting errors for the different trial types in Exp. 1. WM response errors (a, top row) were tightly clustered around high performance ($\sim 0^\circ$ – 10° error), with only very few large errors. The LTM error distribution (b, middle row) did show fat tails (as expected with more guessing), but no clearly discernible differences in tail height (relative to other data features) between the critical experimental conditions.

In sum, given that the present effects were not overly large in size, we believe that a simple continuous error analysis is more appropriate and avoids overfitting in the absence of clear a priori hypotheses regarding differential effects on precision vs. guessing. For transparency, we included the error histograms in the revised Supplement (Supplementary Figure 2). Thank you for encouraging this extension.

11. *It should be noted that, because the LTM test, presents each object from the WM test and asks participants to recall its orientation, this LTM test is really more of a cued associative recall test than a true free recall test. This should probably be clarified for the readers in the intro, results, or discussion so that they don't miss this important detail in the methods.*

We fully agree. Please note that in our manuscript, we did not use the term “free” recall to describe our testing format. We would likewise read ‘free’ recall as freely recalling/listing a set of multiple memory items. We do think that our continuous orientation reporting tests are genuine ‘recall’ tests, in that the probe stimulus provided no information about the original sample orientation. We agree that the test is also associative and ‘cued’ (via the probe object), which we think holds for every ‘recall’ test in one way or another. Like the reviewer, we would find it inaccurate to refer to our tests as “free” recall, but we find it correct to refer to it as ‘recall’.

12. *The “pruning” control analyses in which items were matched on WM performance to equate the degree of error between the conditions and conditionalizing LTM recall on initial recall*

performance is a nice control analysis. Conditionalizing subsequent LTM recall on WM accuracy has precedence in this literature and should be seen as a strength of the manuscript. Readers and reviewers don't always understand the importance of such control analyses, so the hope of this reviewer is that other reviewers or editors will not make the authors remove these control analyses from the manuscript. The results strengthen confidence in the results.

We fully agree and thank the reviewer for these kind words of support.

13. *The control analysis on p. 11 lines 372-377 seems unnecessary. It would be better and more interesting to assess the degree of bias in recall from the other item's orientation on that trial. These results could help inform the somewhat mixed literature on serial dependence effects on WM and LTM (Bae & Luck, 2017, APP, 2019 Psych Science; Rhilinger, Xu, & Rose, 2023, APP; Bae, 2024, APP).*

Thank you, we addressed the question of potential biases between items in our previous replies (points 9-10). We do think that our brief inspection of potential bias toward the orientation of the WM probe is informative. The WM probe in our paradigm showed the same stimulus object (just differently oriented) as was later used as LTM cue, which made it a strong candidate a priori for potential crosstalk or bias.

14. *Finally, the authors appeal to the notion of "retrieval effort" as a critical process that contributes to the benefit to subsequent LTM. One suggestion is that, rather than just ascribing the benefit to effort, an alternative mechanism is the notion of a more elaborative form of retrieval being involved in conducting a search of memory to recover deprioritized items by self-generating retrieval cues (as the authors suggest with the reference to the generation effect). The critical distinction is that having to search memory and generate effective retrieval cues, which is more likely to involve a deeper, conceptual level of processing than simply reporting a prioritized item that was continuously maintained in a high level of activation (in the "focus of attention"), which is more likely to involve a shallower, more perceptual than conceptual level of processing, e.g., Craik & Jacoby's 1975 process view of short-term retention: [https://www.larryjacoby.ca/images/craik %26 jacoby \(1975\).pdf](https://www.larryjacoby.ca/images/craik%26jacoby(1975).pdf)*

We find this a theoretically intriguing possibility. While our experiments were not designed to distinguish different levels of processing, we sympathize with the possibility that LTM benefits may have arisen from generating richer retrieval cues and effectively promoting 'deeper' processing through continuous reporting. We slightly extended our discussion and included a reference to (F. I. Craik & Lockhart, 1972; K. J. W. Craik, 1967) accordingly (on page 25). Thank you very much!

Reviewer 2:

The authors aimed to demonstrate how the prioritization of contents in working memory (WM) impacts long-term memory (LTM) retention. In the visual WM task, one or two items (i.e., one- or two-sample conditions) were encoded one-by-one and maintained in WM during a delay period until the angle of the items was tested at the end of each trial with self-generated angle responses using continuous measures (Exp. 1, 2) or a 2-choice response (Exp. 3). The same items were then tested for their

presented angles in the subsequent LTM test with the same continuous measures. The prioritization in WM was defined in the 2-sample condition based on either WM testing order or retro-cue, in which first tested or cued items became prioritized, and thus the remaining items were deprioritized, and a subset of trials was not tested in the WM probe task to serve as a baseline. The main findings showed that attentional deprioritization reduced immediate WM recall accuracy, while it increased subsequent LTM recall performance.

While the objective of the study - understanding how different prioritization states of contents in WM impact their LTM retentions - is important, and the results are interesting, I still have a few concerns and suggestions.

Major points:

1. My major concern is that the framework is not clear enough to understand either the objectives or the hypotheses of the study. Although the authors introduced the McCabe effect as a framework—explaining how less accessible WM contents due to distractors were better retrieved in LTM—there are still gaps in the establishment of hypotheses for visual WM prioritization. This lack of a clear framework also makes it difficult to identify the study's objectives. Clarifying these objectives could help us better understand how this research contributes to further insights into WM prioritization effects on LTM. For example, WM deprioritization has been studied within the “activity-silence state” framework, demonstrated using similar WM task paradigms (Lorenc et al., 2021; Mallett & Lewis-Peacock, 2019; Rose et al., 2016; Wan et al., 2024). The framework posits that deprioritized WM representations enter a silence state in which they are deactivated but still accessible in WM. Such a state makes these representations less vulnerable to memory updates, which could result from neural interference caused by perceptual inputs or memory competitions, and ultimately improves LTM compared to prioritized WM contents.

We appreciate the reviewer's comment. Our study builds on the well-established empirical finding of testing effects in the LTM literature (e.g., Rowland, 2014), which shows that actively retrieving (or generating) information enhances its long-term retention. While this phenomenon has been extensively studied behaviourally in the context of LTM retrieval, less is known about whether similar effects occur when information is retrieved from WM (“WM-tested”). Our present study set out to examine this, also taking into account previous findings from the WM literature (such as the McCabe effect, and partly mixed findings about the LTM consequences of WM-de/prioritization more generally, see Introduction).

The activity-silent WM framework is an influential mechanistic account on the neural level, which suggests that WM storage is implemented via short-term synaptic plasticity (STSP: Mongillo et al., 2018; Stokes, 2015) instead of persistent neural firing. Whether unattended or “latent” WM storage is implemented in “activity silent” synaptic engrams or alternatively, recruits LTM-like neural processes remains an open debate. Our present work, which is purely behavioural, cannot speak directly to the debate about how unattended WM storage is mechanistically implemented on the neural level. To our knowledge, the activity-silent framework (Mongillo et al., 2018; Stokes, 2015) makes no specific predictions about the potential long-term memory storage of WM contents. We therefore did not consider activity-silent STSP an obvious candidate framework for our present study a priori. That said, in response to the reviewers' suggestion, we now explicitly acknowledge both activity-silent accounts and representational transformations (see below) in our discussion as possible scenarios underlying unattended WM storage on the neuronal level. Whether these frameworks could potentially explain parts of the LTM benefits we are seeing will hopefully be addressed in future work involving neural recordings (please see p. 26 of our updated manuscript).

More recently, the “representational transformation” account has been proposed, supported by neural and simulation evidence. This account suggests that deprioritized representations are transformed into orthogonal representational structures or projected into a different neural space (Piwek et al., 2023; Wan et al., 2022; Yu & Postle, 2021). The representational transformation account is further connected to the primacy effect, in which the initially encoded “sample 1” might be represented in a neural space parallel to the LTM representational space, while “sample 2” might occupy a neural space orthogonal to the original. Hypothetically, the initially encoded representation is supported by original representational structures in LTM, though this has only been demonstrated through model simulations.

The reviewer raises an astute point about possible differences in neural representational geometry between prioritized vs. deprioritized WM contents (Piwek et al., 2023; Yu & Postle, 2021; see also Panichello & Buschman, 2021). At the same time, we wish to exert caution in interpreting our findings along these lines, as to our knowledge, these fascinating recent works made no specific predictions about WM recall (retrieval) and/or about subsequent LTM. In this context, we find it important to highlight that our results suggest no LTM benefit of WM deprioritization *per se* (see p. 14 | 20, and Discussion), but specifically of active retrieval/reporting *after* deprioritization. As regards the primacy effects we observed, an equally plausible interpretation could be (in line with other studies) that they stem from prioritized encoding of the first sample in the trial, rather than from its deprioritization during sample 2 processing (e.g., Sederberg et al., 2006). Thus, while we find the reviewer’s suggestion intriguing, we wish to exert caution in interpreting our LTM results in terms of the WM content’s potential neural representational reformatting. We do think that investigating such potential links with subsequent LTM will be an intriguing avenue for future neuroscientific and modeling work.

2. Additionally, the McCabe effect appears fundamentally different from the WM deprioritization effect at face glance. The McCabe effect is explained by repeated reactivation of WM contents (i.e., covert retrieval) as a mechanism to protect WM targets against distractors, whereas deprioritization promotes WM deactivation. The potential underlying mechanisms that WM deprioritization might share with the McCabe effect should be clearly introduced.

Related to our reply to the previous point, we wish to clarify that our results do not suggest that WM deprioritization alone leads to improved LTM retention (see p 20 | 21, page 24 of discussion). Instead, it appears to be the overt **retrieval** of deprioritized WM contents that leads to LTM benefits, potentially much like the covert retrieval processes proposed in McCabe et al.’s studies. The main distinction, besides differences in task and materials, is that we directly test retrieval effects using overt WM tests, whereas McCabe’s study presumed covert retrieval processes to occur during maintenance. Rather than positioning our findings in opposition to the McCabe effect, we view them as complementary: both studies suggest that retrieval—whether overt or covert—plays a role in strengthening WM contents for subsequent LTM, particularly after deprioritization. Our findings further suggest a contribution of ‘generation’ effects, which could not be assessed in McCabe’s original experimental designs, as well as a role of WM test format. We do hope it is sufficiently explained throughout the revised manuscript that the key factor in our study (like in McCabe’s) is actually **retrieval** from the respective WM states, rather than only storage in these states.

3. In Exp. 1, it is worth considering whether the worse LTM performance for the first tested items (WM Test-1, prioritized items) could be attributed to retroactive interference from Test-2 item retrieval rather than the WM prioritization factor. To control this possible confound, the LTM data for WM Test-1 trials in the two-sample condition could be analyzed based on the presence or absence of the second WM probe test. Furthermore, based on the retroactive interference account, it is possible that LTM for Test-1 items was biased toward Test-2 items when WM Test-2 was present (Zhang & Lewis-Peacock, 2022). If such interference exists, it should be appropriately controlled, for example, by covarying it out in the analysis.

We agree that retroactive interference from the second WM test could be considered a potential confound in interpreting the reduced performance we see in Test 1 vs Test 2. To address this possibility, we report a control analysis in which we directly compared the performance on Test 1 as a function of the presence of a second test (Test 2). (see Footnote 2 on p. 14 of the manuscript. The analysis showed no difference in LTM performance between Test 1 items that were followed by a second test and those that were not [$t(186) = 0.13$, $p = 0.90$], which speaks against a recency- or interference-based explanation of the higher LTM performance we observed for Test-2 WM items.

4. While I appreciate the innovative WM task paradigm, it remains unclear how these modified components contribute to a novel or deeper understanding of WM prioritization.

Thank you – we extended the introduction paragraph on p. 3 | 4 to better outline the key features of our study's task design, compared to more classic designs that examined WM-LTM interactions, e.g., with word lists (e.g., McCabe, 2008) (e.g., McCabe, 2008). Specifically, our object-orientation task enabled us to use a continuous reporting WM test (Exp 1) with which we could also examine subsequent LTM bias to participants' own (subjective) WM report (i.e. "generation"-like effects). Further, the task allowed us to use classic manipulations such as retro-cueing (Exp. 2), which are commonly used in studies of attentional (de)prioritization in visual working memory. Our base paradigm is actually not unconventional, given that stimulus orientation is a common feature used in very many visual WM studies. Our main modification was to present the orientation information with a unique 'carrier' object which allowed us to uniquely probe the associated orientation information in subsequent LTM testing. This would not be possible with more traditional orientation stimuli such as Gabor gratings. Aside from this, we would argue that our tasks are variants of a relatively common visual WM task setup, which we believe provides a valuable complement to the existing literature on WM-LTM interactions with verbal materials.

Minor points

1. There are several suggestions regarding the figures.
 - a. The arrangement of the plots emphasizes the primacy effects by highlighting the differences between sample 1 and sample 2. For Fig. 2, it would be easier to interpret the results if the two-sample data were grouped by encoding order (i.e., sample 1 vs. sample 2) first, with each group including WM Test 1, WM Test 2, and Not Probed (NP) conditions. This arrangement would also allow for clearer comparisons against the NP baseline trials. A similar arrangement could be applied to Fig. 3, with the data grouped by encoding order (sample 1 vs. sample 2), WM probe status (probed vs. not probed), and retrocue status (cued vs. uncued) in sequence. This structure would better emphasize WM prioritization differences, particularly the contrast between cued and uncued conditions.

Thank you for these thoughtful suggestions.. In response, we explored reordering the figure and regrouping the data by encoding order (i.e., WM sample 1 and 2) as proposed. While we agree that

such a layout may facilitate certain comparisons, it comes at the cost of making other aspects of the data very difficult to discern, such as the systematic effects on primacy (3-way interaction) in Exp. 1, which is why we ultimately decided to retain the original structure. That said, to address the reviewer's very valid point regarding fast-forward interpretability of our plots, we added mean lines (in yellow) to the results figures to better visualize the Test 1/2 comparisons across the WM and LTM tasks. We thank the reviewer again for the productive criticism of our plot design.

b. The current color coding for the plots is not very effective. I recommend using warm colors for the prioritized condition and cool colors for the deprioritized condition. Reduced saturations or transparency within the same color scheme could be applied to represent the not-probed conditions. Although higher transparency was applied for LTM compared to WM results, I do not think using the same colors for WM and LTM leads to confusion. Instead, transparency variations could be used to distinguish between WM probed and not-probed results. Additionally, applying the same color scheme to the WM task diagram in Fig. 1, along with a legend indicating the colors for prioritized and deprioritized conditions, would make the prioritization manipulation easier to understand.

Thank you for the feedback regarding the color scheme of our figures. We revised the color mapping to better reflect the WM priority conditions. We now use green color tones for items that were prioritized in WM and pink tones for items that were deprioritized, aligning with the reviewer's suggestion to differentiate conditions using warm and cool colors. We also updated the not-probed condition with distinct color variations to enhance visual separation from the probed (tested) WM samples. We also incorporated the revised color scheme into the WM task diagram (please see Fig. 1), as suggested by the reviewer. Thank you very much.

c. The upper bars across conditions for the significance notations are unclear, as it is not evident whether they represent interactions or main effects.

Thank you, we agree and added clarification in the Figure captions.

d. In Fig. 2c, the plot showing the LTM-recall bias toward the WM report (i.e., similarity) is difficult to interpret. It would be clearer if the distributions of the differences between the memory report (i.e., either WM or LTM) and the original samples (adjusted to a 0-point baseline), were plotted in separate sub-figures. This would allow for a better understanding of whether the LTM recall was biased toward the WM report or the original samples.

We assume that the reviewer refers to classic plots of the error distribution relative to the physical target and to the WM report, respectively. In the revision, we included such plots in the supplementary material (please see our response to reviewer 1, comments number 10-11 and Supplementary Figure 2) for the reviewers to verify. The histograms provide no visually salient information beyond that reported in the paper, i.e., that the LTM "error" (deviation) relative to the WM report was generally smaller than that relative to the original sample orientation.

2. There is a typo on page 14 (464): Fig. 1c seems to refer to Fig. 3c instead.

Thank you, the typo is now corrected.

Reviewer 3:

This manuscript comprises three experiments designed to test the impact of deprioritization and retrieval effort in working memory (WM) on later retrieval from episodic long-term memory (LTM). Specifically, participants completed an online study where trials of one or two objects presented in different orientations were followed by immediate WM tests and a surprise LTM test. The results of Experiment 1 showed that the error in reproducing the orientation of the objects from WM was unsurprisingly greater when participants were tested on two objects than one object, but the reverse was true in LTM: Reproduction error was reduced for objects that were tested second (and thus temporarily deprioritized) versus first. As there was also a primacy effect, Experiment 2 used retro-cues to attempt to disentangle the impact of prioritization and position in the trial, which indeed showed a greater LTM benefit of reduced reproduction error for uncued (i.e., deprioritized) objects versus cued objects. Finally, Experiment 3 showed that these effects are subject to the nature of the WM test, such that the key results of Experiment 1 were nullified when using a simple recognition test instead of reproduction. The authors interpret the findings as consistent with a retrieval effort explanation of the importance of WM for LTM.

Evaluation and major comments:

This was an interesting set of experiments that was straightforward to follow and well situated in the literature on this important WM/LTM topic. Most of my constructive comments further below are points of clarification, but the potentially trickier ones regard the key results on which the major theoretical interpretations hinge.

There are a few points on this issue of interpretation. First, it made sense to use retro-cues in Experiment 2 to follow up Experiment 1, but the key interaction between WM testing and cueing (lines 477-479) is an ordinal one, which are known to be difficult to unambiguously interpret (Loftus, 1978; Wagenmakers et al., 2012). In other words, it is possible this interaction may be removable when transforming the data to another scale that does not have the same potential psychological ceiling/floor as reproduction error. Indeed, key interactions in this WM/LTM topic have been transformed away in previous work (e.g., Labaronne et al., 2023), suggesting that one should be cautious to interpret interactions that may be exacerbated by mere issues of scale. So, my advice here would be to transform the data to another scale to ensure that this key interaction still occurs and is not removable.

Thank you for this important advice. In the revision, we re-ran the 2 x 2 ANOVA analysis testing for this critical interaction effect (Testing x Cueing) after logit-transforming the LTM performance data to account for potential scaling issues (as suggested in Wagenmakers et al., 2012; Labaronne et al., 2023). Importantly, the interaction effect remained significant [$F(1,88) = 4.253$, $p = 0.042$, $\eta^2 = 0.005$]. We included this additional information, which corroborates the robustness of our findings, in the revised supplementary information (Suppl. Analysis 1). Thank you very much for this suggestion.

The second issue is that quite a lot of heavy lifting in the interpretations rests on null p-values, which also cannot be taken as evidence for the null. Is there any indication of how strong the evidence is for null in these situations, e.g., by using a Bayesian analytic approach? For example, the fact that the key results of Experiment 1 are nullified in Experiment 3 is interesting but could be an issue of power rather than a true null effect (and I have another question about power further on).

We fully agree with the reviewer that null results should be interpreted with caution. However, we wish to clarify that our conclusions about the role of WM-testing format do not rely solely on a null finding. Specifically, we report a mixed-effects ANOVA directly comparing the WM-testing effects found in Exp.

1 vs. Exp. 3 (see p. 22 in our manuscript). This analysis shows that the WM-testing effect in Experiment 3 is in fact **significantly reduced compared to Experiment 1** [$F(1,292) = 17.755, p < 0.001, \eta^2 = 0.013$]. In other words, our finding of a reduced WM-testing effect with a more recognition-like WM-test format is not only supported by a null result (within Exp. 3), but also by a statistically significant difference compared to WM testing with continuous reporting (between experiments). We consider this an important result, given that aside from the different WM-test procedures used, the designs of Experiments 1 and 3 were practically identical, which allows for a direct comparison of the LTM-test data between the two experiments – although we are also aware of potential limitations with this approach (see below).

Finally, related to this specific finding of nullified results in Experiment 3, I think that the conclusion that overt retrieval effort is interesting but premature when considering the design. Specifically, the method of WM testing is not the only thing that has changed about the study, but perhaps also the participants' approach to the task. If I am asked to make a simple recognition choice based on the content of my WM, I am probably less likely to try to actively encode and maintain as much detail about the orientation of the stimuli compared to a reproduction test as in Experiment 3. Thus, it is currently impossible to know if the null results occurred due to the obvious difference in retrieval demands (as the authors propose) or whether the participants were less proactive in their active maintenance of such detailed information that only became relevant much later in the surprise LTM test. The issue of task engagement has been recently identified as a potentially important moderating factor of the McCabe effect (Cotton et al., 2024), and I myself have tried to disentangle retrieval demands versus task approach in Experiment 3 of the paper cited in this manuscript (Loaiza et al., 2021). It is not immediately obvious to me how to address this issue without conducting another experiment that clearly disentangles encoding/maintenance effort versus overt retrieval effort.

We appreciate this comment and agree that potential differences in retrieval demands and participants' encoding or maintenance strategies may play a role and can be considered a possible limitation of our study. We now mention this more explicitly in the revised discussion (please see p. 25). At the same time, we find it important to emphasize that the WM task in Exp. 3 was not a simple recognition task, but a delayed **comparison** task (clockwise/counterclockwise rotational judgment relative to the probe orientation). Unlike standard recognition (old/new, or same/different), our task in Exp. 3 did require participants to explicitly encode and maintain the stimuli's orientation in rotational coordinates, only without the self-generating continuous recall that was required in Exp. 1 (see also our response to Reviewer 1).

While this distinction doesn't fully rule out the possibility that the change in test format might also have led to differences in encoding or maintenance, we believe that our choice of test format in Exp. 3 reflects a good compromise to narrow down to differences in retrieval requirements. We hope that future work, potentially also incorporating neuroscientific approaches, will help to further disentangle the effects of retrieval demands, encoding, and maintenance strategies in greater detail (see Discussion p. 26).

Signed,

Vanessa Loaiza

(Please note that I only sign reviews when the work is obviously tied to my own)

More minor points of clarification:

- Line 128: Could you clarify what you mean by "negative results"?

We have rephrased this sentence to clarify. Please see p. 3:

“On the other hand, the abovementioned ‘McCabe effect’ on subsequent LTM has in a few studies also been observed in trials where overt WM testing was omitted (Loaiza et al., 2021; McCabe, 2008)”.

- Lines 136-139: I do not understand this sentence for two reasons. First, I do not understand what is meant by “no differential LTM benefit” (perhaps different benefit depending on retrieval method?). Second, the second part of the sentence does not seem to follow the first – why is “maintain[ing] visual information” inconsistent or “unlike” these previous studies’ lack of “differential LTM benefit”?

Thank you for pointing this out. With “no differential LTM benefit,” we indeed meant to refer to the absence of different LTM outcomes depending on the type of retrieval method. Further, we wanted to outline that previous research on the McCabe effect mostly used verbal stimuli, whereas our study focused on a more “visual WM” setting. To improve clarity, we have revised the phrasing in the introduction of the manuscript (p .3) to read:

“While previous studies found no effect of overt WM testing on the ‘McCabe’ effect with word lists (Loaiza et al., 2021; McCabe, 2008), our WM tasks required participants to maintain visual information, specifically, the orientations of one or two rotated objects.”

- Lines 139-141: Please provide citations for the “previous studies” in the first part of this sentence.

Apologies for the oversight; we have now added these citations, specifically (e.g., LaRocque et al., 2015; Jeanneret et al., 2023; Wang & Van Ede, 2024).

- Lines 144-148: I did not understand this sentence until later in the paper. I would recommend giving a bit more methodological information earlier on or adjusting it some other way so that the reader does not have to read ahead to understand it.

We thank the reviewer for pointing this out. We have added further details on the continuous orientation recall test, which is central to our visual WM paradigm (see revised text on p 3 | 4).

- Could you clarify the rationale for the sample sizes of the experiments? They seem quite high compared to previous research, which is not a bad thing, but if some a priori power analysis was conducted to justify greater sample sizes that would be good to know. The sample size is also less for Experiment 2 compared to the other experiments, so it would be good to know why this was the case.

The relatively large sample sizes reflect our intention to ensure sufficient power despite the complexity of the tasks and the online testing format, which we also had little experience with for this type of paradigm, and we assumed that the online setting could lead to relatively high exclusion rates. Other online experiments (e.g., Cotton et al., 2024) collected similarly large samples, even in more homogeneous samples like a group of young psychology students. The smaller sample size in Exp. 2 was based on the assumption that the retro-cue manipulation (as a more established WM paradigm) would yield stronger effects than we saw in interim data checks. In hindsight, this was likely too optimistic, which is also reflected in the mid-experiment adjustment of cue validity (after inspecting the preliminary WM task data, we increased the cue validity from 75% to 83.33%; see p. 7 Methods of Exp. 2) to strengthen the manipulation and encourage participants to more strongly rely on (or “use”) the retro-cue.

- *Figure 3d: I assume that these results are collapsed across sample position, but it would be good to break it down by position just to visually verify that effect is indeed stronger for uncued items for both primacy and recency positions.*

We now provide a plot as requested by the reviewer as a Supplementary Figure (Suppl. Fig. 3, also shown below). The WM-testing benefit for uncued items is found to be similarly present for the first and second sample (i.e., both primacy and recency positions). This mirrors our results report in the manuscript that no interaction with sample position was found (p. 19).

- Please report effect sizes alongside the simple effects/t-tests (e.g., Cohen's *d*).

Thank you, we now included Cohen's *d* for all t-tests, and η^2 for all ANOVAs, throughout the manuscript.

- Lines 639-642: *Harkening back to the previous major issue on interpretation, it would be good to spell out exactly why the effect is better explained by overt versus covert retrieval before jumping to the next point.*

Thank you for this comment. We would like to clarify that we did not intend to argue that covert retrieval would play no role in our study (particularly in Exp. 1, see Discussion p. 23), but that independently (or additionally), LTM benefits after deprioritization may also arise from overt retrieval. Specifically, we think that this interpretation is supported by (i) the finding of an effect in Exp 2 (where an explanation in terms of covert retrieval appears unlikely, p. 19), (ii) increased bias to the WM report (away from the original sample), which is not easily attributed to canonical biases in WM and LTM reports (please see also our detailed responses to Rev 1 and extended Supplementary Materials), and (iii) significantly lower LTM benefits in Exp. 3 (despite the 2nd WM delay providing additional covert retrieval opportunities similar to Exp. 1). We hope these points become clearer in our revised manuscript (see discussion).

- Lines 885-900: *I found the explanation of the pruning method vague. I would recommend giving more detail, and ideally also include your analysis script along with your data so that other researchers can reproduce the method in their own work.*

We thank the reviewer for this helpful suggestion. We have now extended the description of the pruning method in the manuscript to clarify the procedure (see p. 8, section "Pruning for equivalent WM performance"). In the process, we noticed a minor difference between the pruning approaches used in Exp. 1 and 2: In Exp. 1, pruning was based on each participant's best-performing WM condition, whereas in Exp. 2, the mean WM performance across all conditions was used as the pruning target. To ensure consistency, in the revision, we use the latter approach also in Exp. 1 and updated the results accordingly in the manuscript (see p. 15; note minor changes in numerical values, but with identical statistical results patterns and conclusions). The pruning and analysis scripts, which have already been publicly available alongside the preprint at the time of our first submission

[\[10.5281/zenodo.13867138\]](https://doi.org/10.5281/zenodo.13867138), have been updated to reflect these changes.

We are grateful to the reviewers for their very helpful comments and suggestions.

Response to reviews (COMMSPSYCHOL-24-0709A / R2)

Reviewer 1:

The authors have generally done a nice job revising their manuscript. Only a couple of minor issues remain.

They state (p. 2, starting line 98) "While many of these studies found that prioritization IMPROVED subsequent LTM (Fan & Turk-Browne, 2013; Jeanneret et al., 2023; LaRocque et al., 2015; Reaves et al., 2016; Strunk et al., 2019; Wang & Van Ede, 2024), others found no such effect (Bartsch et al., 2018; Mao Chao et al., 2023) or even found superior LTM when attention was diverted from the WM information (Rose et al., 2014)." However, as they correctly state later on (p. 25), "A previous study found no evidence that unattended WM maintenance would improve subsequent LTM (LaRocque et al., 2015)..."

The first sentence should move the LaRocque et al., 2015 reference to the Bartsch et al., 2018 and Chao et al., 2023) references. Also note that the Mao is Chao's first name. The Reference is Chao, Xu, Loaiza, & Rose.

This is a nice contribution to the literature. I look forward to seeing forthcoming work on these interesting and important topics.

We thank the reviewer for their positive evaluation of our revised manuscript and for pointing out the inconsistency regarding the interpretation of LaRocque et al. (2015). We fully agree with the reviewer's assessment and have updated our revised manuscript accordingly. We have also corrected the Chao et al. reference to use the first author's surname.

Reviewer 3:

Thanks to the authors for addressing my comments. I have read through the revised manuscript, and I think that it satisfactorily addresses these issues that I and the other reviewers raised. I have no further comments, and I recommend publication.

We sincerely thank the reviewer for their careful reassessment of our revised manuscript and for their positive recommendation.